# Effect of agricultural extension on technology adoption by Palestinian farmers under Israeli occupation in the West Bank

**Nakamura Tomoki**[1]*, **Kashiwagi Kenichi**[2], **Ujiie Kiyokazu**[3]

**1** Graduate School of Life and Earth Sciences, University of Tsukuba, Tsukuba, Ibaraki, Japan, **2** Faculty of Humanities and Social Sciences, University of Tsukuba, Tsukuba, Ibaraki, Japan, **3** Faculty of Life and Environmental Sciences, University of Tsukuba, Tsukuba, Ibaraki, Japan

* s2030240@s.tsukuba.ac.jp

**Data Availability Statement:** All relevant data are within the paper.

**Funding:** The authors received no specific funding for this work.

## Abstract

Even during the conflict, agricultural extension by the Palestinian Authority has played an important role in agricultural development in the West Bank of the Occupied Palestinian Territories (OPT). The Ministry of Agriculture of the Palestinian Authority provided the necessary agricultural extension services for Palestinian farmers affected by the Israeli settlements and Segregation Wall. Despite such importance of agricultural extension, few quantitative studies have examined its effect on Palestinian farmers. Therefore, the objective of this study was to quantify the effect of agricultural extension on technology adoption by Palestinian farmers for appropriate evaluation of the agricultural policies by the Palestinian Authority. The microdata of 79,446 agricultural holdings from the Agricultural Census 2010, which was the only microdata officially published and available at the time of this study, was used. Then, the Propensity Score Matching (PSM) method was employed to mitigate the endogenous bias caused by self-selection by farmers in receiving the agricultural extension. The results showed that agricultural extension has positively and significantly affected the adoption of five technologies, namely improved crop varieties, chemical fertilizers, organic fertilizers, pesticides, and biological control. The estimated increase in the adoption rate of those technologies as the average treatment effects on the treated (ATT) by the nearest-neighbor matching method were by 7.1, 7.7, 5.4, 6.8, and 3.8 percentage points respectively. This study proved that agricultural extension promoted the adoption of those technologies even in the conflict. Therefore, agricultural extension by the Palestinian Authority plays an important role in farming by Palestinian farmers. In order to maintain and improve farmers' livelihoods sustainably, it is necessary to continue the agricultural extension by the Palestinian Authority in the future, considering the behavior of farmers.

## Section 1: Introduction

Conflict is directly associated with poverty, hunger, and agricultural development, thus it is essential to understand that gains in agricultural productivity and food security improve

**Competing interests:** The authors have declared that no competing interests exist.

poverty and hunger indicators [1]. The effects of conflict are diverse, and the same conflict can have many different effects on different people across time and space [2]. In the past decade, microeconomics of violent conflict has been identified as a new subfield of empirical development economics, and future studies are expected to contribute to peacebuilding, the development of postconflict institutions, the behavior of economic actors in conflict areas, and the role of emotions in decision-making [3]. In a study on the effects of conflict in Colombia, negative income shocks caused by the conflict had substantial impacts on agricultural production and welfare levels, forcing many poor households to resort to migration [4]. Another empirical analysis using panel data collected from Palestinian households before and after the 2014 Israeli war on the Gaza Strip showed that income destabilization resulting from the conflict reduced long-term household resilience [2]. Under such influence of the conflict, agricultural extension plays a fundamental role in postconflict agricultural development [5].

The West Bank of the Palestinian territories has been under Israeli occupation since 1967. Under such an environment, Palestinian farmers continue to farm for their livelihood. The West Bank was selected as the target area for this study because assistance to such farmers was recognized as an urgent issue. Some parts of the agricultural land in the West Bank were confiscated by the Israeli Occupation Authorities for expanding Israeli settlements and constructing the Israeli Segregation Wall during the conflict [6]. In the affected areas, Palestinian farmers have limited access to agricultural land and water, thus reducing agricultural productivity [7]. Under such a situation, the Ministry of Agriculture of the Palestinian Authority set rehabilitation of agriculture affected by the conflict as a strategic objective in the National Agriculture Sector Strategy and provided necessary agricultural extension services to farmers affected by the Israeli occupation. Thus, the Palestinian Authority disseminates agricultural technologies and provide agricultural inputs to those affected farmers as public support. Despite such importance attached to agricultural extension, few quantitative studies have examined how agricultural extension affects technology adoption by Palestinian farmers.

This study examines the effect of the agricultural extension by the Palestinian Authority on technology adoption by Palestinian farmers in the West Bank. Restrictions on agricultural land use due to the construction of the Israeli Segregation Wall and Israeli settlements would have an impact on agricultural activities. Especially in Area C, confiscation and destruction of agricultural land frequently occur [6]. Therefore, it was hypothesized that farmers affected by the conflict are more likely to receive agricultural extension and adopt technologies to mitigate the impacts even under such challenging conditions.

Several studies have examined the impact of agricultural extension under the influence of conflict. A study on agricultural extension in Iraq during the 2003–2012 conflict describes the effectiveness of providing agricultural inputs to farmers, disseminating agricultural technologies, and training farmers. It also notes that agricultural extension in the conflict included special challenges different from normal times [8]. A study on the impact of the conflict in northern Sri Lanka that ended in 2009 found that agricultural extension was effective in rebuilding and peacebuilding local communities damaged by the conflict, and the impact of the conflict on women farmers and smallholder farmers was particularly large [9]. According to a study on the reconstruction of agricultural extension under conflict-affected conditions in Afghanistan, it was necessary to strengthen the extension officers' capacity; increase the number of female officers; develop and disseminate improved crop varieties; and organize meetings, cultivation demonstrations, and field study tours [10].

The rest of the sections of this study are presented as follows: Section 2 describes agricultural extension under Israeli occupation, Section 3 explains the analytical framework, and Section 4 shows data and descriptive statistics. Section 5 discusses the results of this study, and Sections 6 and 7 provide discussion and conclusions, respectively.

## Section 2: Agricultural extension under Israeli occupation

According to the Oslo II Accord officially agreed in 1995, 62.9% of the area in the West Bank is classified as Area C under Israeli control. In contrast, 18.8% is classified as Area B under the Israeli–Palestinian administrative control and security, and 18.3% is classified as Area A under full Palestinian control [11]. In Area C, some parts of the agricultural land have been confiscated by the Israeli Occupation Authorities for the expansion of Israeli settlements, construction of the Israeli Segregation Wall, and bypass roads. Consequently, the agricultural land area in the West Bank has gradually decreased, and some parts have become inaccessible to Palestinian farmers [11].

The construction of the Israeli Segregation Wall has continued since 2002. Approximately 85% of the Israeli Segregation Wall was built on the Palestinian side of the internationally recognized Green Line (the 1949 Armistice Line). The wall grabs fertile agricultural land, isolates Palestinian communities in enclaves undermining the territorial contiguity between Palestinian villages, and limits access to natural resources causing water shortages [12, 13]. As of 2021, the Israeli Segregation Wall was 712 km long. Approximately 9,000 dunum, equivalent to 900 hectares, of irrigated agricultural land has been confiscated for constructing the Israeli Segregation Wall [7]. Additionally, approximately 10% of land in the West Bank is isolated as "seam zones" between the Israeli Segregation Wall and the Green Line [6]. According to UNCTAD [7], about 11,000 Palestinians, including farmers, live in the seam zone and those farmers holding agricultural lands in the zone must obtain special permits from the Israeli Occupation Authorities to access their farmlands.

According to B'tselem [14], the construction of Israeli settlements began after the Third Middle East War in 1967. By early 2021, over 280 settlements had been built in the West Bank, including East Jerusalem. The Israeli settler population in the West Bank had reportedly increased from 198,315 in 2000 to 311,136 in 2010 and exceeded 650,000 in early 2021. Moreover, the Israel Authorities promote the construction of outpost settlements and encourage Israeli settlers to conduct agricultural activity there, confiscating farmland and pastures from Palestinian farmers. Consequently, it was estimated that about one-third of the farmland in the West Bank had decreased [15]. These settlements discharge wastewater, affecting surrounding farmlands and irrigation water [16].

Constraints on the use of water resources by the Israeli Occupation Authorities are one reason for the shortage of irrigation water in the West Bank. In 2007–2008, a 35–40% reduction in the production of rainfed crops such as wheat, forage, olives, grapes, and other fruit trees was observed due to the shortage of irrigation water [17]. By comparing the Palestinian territories with Israel and Jordan, which have similar natural environments, some studies have estimated the influence of the conflict on agricultural production in the Palestinian territories. According to UNCTAD [6], the crop yield was approximately 50% of Jordan's and about 43% of Israel's. The fruit yield in the West Bank was about 53%, field crop yield was about 33% and olive yield was about 36% of Israel's crop yield. Limited access to agricultural land and water, along with poor fertilization, market constraints, and low access to agricultural inputs, had caused those production gaps [6].

According to the 2010 Agricultural Census by the Palestinian Central Bureau of Statistics (PCBS), West Bank had 90,908 agricultural holdings [18]. The Census also revealed that most agricultural holdings were considered small holdings of less than 10 dunums (equivalent to one hectare) owing to fragmentation and division because of inheritance and confiscation by the Israeli Occupation Authorities [11]. Agricultural lands located in Area C are particularly vulnerable, and most lands are always facing confiscation by the Israeli Occupation Authorities.

In such a situation, Palestinian National Agriculture Sector Strategy (NASS) 2014–2016 stipulated rehabilitation of agriculture as one of its strategic objectives. As a result of the strategy, rehabilitation was undertaken mainly in Area C along with providing emergency support to hundreds of households whose homes, farms, barns, and income sources were affected by the conflict [19]. The subsequent agricultural policy for 2017–2022 also included farmers' empowerment to overcome the negative impacts of the conflict, while strengthening their presence on the land and improving their agriculture, incomes, and standards of living [11]. Palestinian National Agricultural Extension Strategy (PNAES) 2016–2019 also aimed to provide the necessary agricultural extension services to farmers directly affected by the Israeli occupation. The PNAES mentions that farmers are provided the necessary support and strengthened their capacity to continue farming, and are protected from Israeli occupation to enhance food security [20].

Based on these policies, the Ministry of Agriculture set up Directorates of Agriculture in all governorates to provide agricultural extension services to local farmers. It includes technology transfer and supplying farming inputs to help farmers who are facing difficulties owing to the conflict [11]. In the West Bank, extension officers in the Directorates of Agriculture are mainly in charge of the agricultural extension services. The total number of extension officers was 168, of which 123 (73.2%) were male offices and 45 (26.8%) were female officers [21]. The methods of agricultural extension included individual farm visits by extension officers, lectures, demonstrations, Farmer Field Schools (FFSs), and distribution of publications, TV, and radio programs [21]. Farmers were free to participate in those extension activities. Farmers received technical guidance from the extension officers when faced with agricultural issues. In 2009 and 2010 covered by the Agricultural Census 2010, 7,541 field visits by extension officers, 360 extension lectures, 12 FFSs (Farmer Field Schools), and 276 distributions of technical materials were conducted [21]. Therefore, the impact of agricultural extension on technology adoption by Palestinian farmers should be quantified for appropriate evaluation of the agricultural policies by the Palestinian Authority.

## Section 3: Analytical framework

The effect of the agricultural extension on technology adoption by Palestinian farmers was quantified. The identification strategy has two components: ordinary least squares (OLS) on the linear probability model for preliminary estimation and propensity score matching (PSM) [22, 23]. The OLS formula on the linear probability model is as follows:

$$Y_i = \beta_0 + \beta_1 D_i + \boldsymbol{H}_i\prime\boldsymbol{\lambda} + \mu_i, \tag{1}$$

where $Y_i$ denotes the binary variable related to technology adoption by agricultural holdings; $D_i$ is a dummy variable that captures receipt of agricultural extension; $\boldsymbol{H}_i$ represents a vector of the control variable that measures the characteristics of agricultural holdings; coefficients of $\beta_0, \beta_1$, and a vector $\boldsymbol{\lambda}$ are the unknown parameters to be estimated; $\mu_i$ is the error term. The parameters of $\beta_1$ measure the treatment effect of agricultural extension.

The following binary dummy variables as a dependent variable were used, whether the agricultural holdings would adopt technologies of (1) improved crop varieties, (2) chemical fertilizers, (3) organic fertilizers, (4) pesticides, and (5) biological control. A binary dummy variable was used, whether the agricultural holding would receive agricultural extension, as an independent variable. As covariates, distances from the Israeli Segregation Wall and Israeli settlements were included, land classification by the Oslo II Accord as the influence of the conflict, characteristics of agricultural holdings, agricultural lands, agricultural products, and governorates. The selection of the covariates was guided by previous empirical studies on technology

adoption by farmers [24–26]. Knowler et al. [27] analyzed 23 empirical papers on farmers' adoption of agricultural technologies and extracted 170 independent variables that influenced farmers' decision-making. Ali (2021) conducted an empirical analysis of the farmers' determinants of choice of adopted practices toward climatic risks [28]. Those independent variables employed in earlier studies were also selected in this study.

The impact of the Israeli Segregation Wall and the Israeli settlements would have greater physical and mental impacts on the agricultural holdings because of their proximity; therefore, the distances were calculated using geographic information and included as covariates. Agricultural products, categorized as vegetable producer, fruit and olive producer, field crop producer, and mixed farmer with livestock, were used as covariates. The Israeli Occupation Authorities have restricted its use of water resources in much of the West Bank, leading to chronic irrigation water shortages [6]. In general, vegetables require the largest amount of irrigation water for cultivation, followed by field crops, fruit trees, and olives. However, the amount of irrigation available is a limiting factor, making it difficult for field crop farmers and fruit and olive farmers to convert to vegetables. Therefore, these crop categories were treated as covariates representing farmer's characteristics in this study.

OLS estimates include bias caused by unfoundedness. Therefore, the propensity score matching (PSM) method [22, 23] was employed to mitigate the endogenous bias caused by the self-selection in receiving agricultural extension. The PSM method assumes that conditioning on observable variables eliminates sample selection bias [29, 30]. The concept of the PSM method is to match individual holdings that have received agricultural extension (treated group) with those that have not received it with similar observable characteristics (control group). Following Rosenbaum and Rubin [23] and Dehejia and Wahba [31], the average treatment effects on the treated (ATT) were examined as follows:

$$ATT = E(Y_i(1) - Y_i(0)|D_i = 1) = E(Y_i(1)|\mathbf{Z}, D_i = 1) - E(Y_i(0)|\mathbf{Z}, D_i = 1), \quad (2)$$

where $Y_i$ denotes the outcome variables of technology adoption by agricultural holdings that received agricultural extension ($Y_i(1)$) and those that did not ($Y_i(0)$). $\mathbf{Z}$ is a vector of covariates that capture the characteristics of agricultural holdings. $D_i$ is a dummy variable indicating agricultural holdings with agricultural extension ($D_i = 1$) or without ($D_i = 0$). The counterfactual, $E(Y_i(0)|\mathbf{Z}, D_i = 1)$, is not observable, but the propensity score can create a comparable counterfactual situation to match the holdings with agricultural extension. Assuming this conditional independence, ATT is expressed as

$$ATT = E(Y_i(1)|D_i = 1, p(\mathbf{Z})) - E(Y_i(0)|D_i = 0, p(\mathbf{Z})), \quad (3)$$

where $p(\mathbf{Z})$ is the propensity score, i.e., the probability of receiving agricultural extension based on observable characteristics of agricultural holdings. For estimating $p(\mathbf{Z})$, the logit model was used. Single nearest-neighbor, Kernel, and Radius matching methods were used to confirm the robustness of the PSM estimation.

Finally, the mean values of the covariates were compared between the treated and control groups before and after matching by conducting balancing tests to evaluate the reliability. First, a *t*-test was used to compare the mean value of each covariate between the treated and control groups after matching. The mean differences would be insignificant if the matching was successful. Second, the pseudo-*R*-squared was compared before and after the matching, as suggested by Sianesi [32]. A successful matching implies that the pseudo-*R*-squared after the matching is lower than that before the matching. Additionally, Oster's coefficient stability was employed to test the sensitivity of unobservables [33]. All estimates in this study were performed using the statistical analysis application Stata/IC 16.1 developed by STATA Corp LLC.

## Section 4: Data and descriptive statistics

To empirically prove the effect of the agricultural extension, the Agricultural Census 2010 micro-data published by the PCBS was used. At the time this study was conducted, only the microdata of the Agricultural Census 2010 was available. A subsequent agricultural census was conducted in 2021, but no microdata has been published. Geographic data on the Israeli Segregation Wall, Israeli settlements, land classification by the Oslo II Accord, and boundaries of Palestinian localities were obtained from the website of the Humanitarian Data Exchange, United Nations Office for the Coordination of Humanitarian Affairs. The Agricultural Census 2010 was the first to be conducted in the Palestinian territories, while it was conducted from October to November 2010 in the West Bank. The data was collected using questionnaires administered by surveyors through individual visits to agricultural holdings in all governorates. In the Census, the unit of the survey was agricultural holding, implying an economic unit of agricultural production.

The micro data of the Census used in this study were provided by PCBS for pure scientific studies dedicated to research and development purposes under a license agreement for the use of processed micro data. The microdata was composed of 111,310 agricultural holdings; of which, 90,908 were of the West Bank and 20,402 of the Gaza Strip. The impacts of the conflict on agriculture in the Gaza Strip are different from those of the West Bank: repeated airstrikes, blockades causing the movement restriction of people and goods, less accessibility of agricultural inputs, and restricted access to external markets [34, 35]. Therefore, it was determined that it was necessary to establish a specific model for the Gaza Strip different from the West Bank, which includes specific variables reflecting properly these particular factors, and the Gaza Strip could not be included in this study. In the Census, the legal status of holders of agricultural holdings was classified into individual, partnership, household, company, government, society, and others. However, companies, governments, and societies were excluded from this study because they were not directly targeted for agricultural extension by the Palestinian Authority, and their holding areas were much larger than other types of holdings. Thus, there were 90,140 agricultural holdings owned by individuals, partnerships, and households in the West Bank. Furthermore, the types of agricultural holdings were classified into plant (holdings producing crops), animal (holdings producing livestock), and mixed (holdings producing both crops and livestock). This study estimates the effect of agricultural extension on technology adoption in crop production. Therefore, data from 79,446 agricultural holdings of plant and mixed holdings producing crops in the West Bank was used.

The geographic data of the conflict was used as variables by calculating the shortest distance from the localities where each agricultural holding was located to the Israeli Segregation Wall and Israeli settlements using the Geographic Information System (GIS). The Census microdata did not include information on land classification by Oslo II Accord. To identify each locality's land classification, GIS was used to identify and integrated into the microdata to be used as variables. The classification of localities based on the Oslo II Accord means the difference in authorities in charge of administration for each locality, which also affects agricultural extension.

Table 1 summarizes the descriptive statistics of the agricultural holdings in this study. Of the 79,446 agricultural holdings covered by this study, 7,922 received agricultural extension in 2009, accounting for 11.08%. As Table 1 shows, the variables used in this study were classified as the influence of the conflict; characteristics of agricultural holdings, agricultural land, and agricultural products; and the governorates.

## Section 5: Results

### OLS regression

Table 2 shows the results of the OLS regression on the linear probability model for preliminary estimation. According to the results, receiving agricultural extension positively and

**Table 1. Descriptive statistics of agricultural holdings in this study.**

| Variables | Total holdings (n = 79,446) | | Holdings that received agricultural extension (n = 7,922) | | Holdings that did not receive agricultural extension (n = 71,524) | | The difference in the means | |
|---|---|---|---|---|---|---|---|---|
| | Mean | S.D. | Mean | S.D. | Mean | S.D. | | |
| *Adoption of agricultural technologies:* | | | | | | | | |
| Improved crop varieties (1 = Yes) | 0.224 | 0.417 | 0.385 | 0.487 | 0.206 | 0.404 | 0.179 | *** |
| Chemical fertilizers (1 = Yes) | 0.265 | 0.441 | 0.425 | 0.494 | 0.247 | 0.431 | 0.178 | *** |
| Organic fertilizers (1 = Yes) | 0.620 | 0.485 | 0.732 | 0.443 | 0.608 | 0.488 | 0.124 | *** |
| Pesticides (1 = Yes) | 0.447 | 0.497 | 0.601 | 0.490 | 0.430 | 0.495 | 0.171 | *** |
| Biological control (1 = Yes) | 0.159 | 0.365 | 0.232 | 0.422 | 0.150 | 0.358 | 0.082 | *** |
| *Influence of the conflict:* | | | | | | | | |
| Distance from the Israeli Segregation Wall (km) | 5.520 | 4.574 | 5.717 | 5.398 | 5.498 | 4.472 | 0.219 | *** |
| Distance from Israeli Settlements (km) | 3.946 | 3.139 | 4.127 | 3.520 | 3.926 | 3.094 | 0.201 | *** |
| Land Classification by the Oslo II Accord: | | | | | | | | |
| • Area A (1 = Yes) | 0.372 | 0.483 | 0.398 | 0.489 | 0.369 | 0.483 | 0.029 | *** |
| • Area B (1 = Yes) | 0.527 | 0.499 | 0.448 | 0.497 | 0.536 | 0.499 | -0.088 | *** |
| • Area C (1 = Yes) | 0.083 | 0.276 | 0.132 | 0.339 | 0.078 | 0.268 | 0.055 | *** |
| • Israeli Declared East Jerusalem (1 = Yes) | 0.001 | 0.035 | 0.003 | 0.051 | 0.001 | 0.033 | 0.002 | *** |
| • Hebron H1 (1 = Yes) | 0.016 | 0.125 | 0.015 | 0.122 | 0.016 | 0.125 | -0.001 | |
| • Nature Reserve (1 = Yes) | 0.001 | 0.031 | 0.005 | 0.068 | 0.001 | 0.023 | 0.004 | *** |
| *Characteristics of agricultural holdings:* | | | | | | | | |
| Age of holder (years old) | 51.832 | 14.008 | 51.647 | 13.945 | 51.853 | 14.015 | -0.206 | |
| Male holder (1 = Yes) | 0.929 | 0.256 | 0.947 | 0.223 | 0.927 | 0.259 | 0.020 | *** |
| Educational background of the holder: | | | | | | | | |
| • Preparatory or less (1 = Yes) | 0.656 | 0.475 | 0.657 | 0.475 | 0.656 | 0.475 | 0.001 | |
| • Secondary or associate diploma (1 = Yes) | 0.221 | 0.415 | 0.220 | 0.414 | 0.222 | 0.415 | -0.001 | |
| • Bachelor or above (1 = Yes) | 0.122 | 0.328 | 0.123 | 0.328 | 0.122 | 0.328 | 0.000 | |
| Main occupation: Agriculture (1 = Yes) | 0.215 | 0.411 | 0.357 | 0.479 | 0.200 | 0.400 | 0.157 | *** |
| The main purpose of production: For sale (1 = Yes) | 0.186 | 0.389 | 0.339 | 0.473 | 0.170 | 0.375 | 0.169 | *** |
| Management by the holder (1 = Yes) | 0.724 | 0.447 | 0.727 | 0.446 | 0.724 | 0.447 | 0.003 | |
| Total members of a holding (persons) | 6.209 | 2.799 | 6.384 | 2.864 | 6.189 | 2.791 | 0.195 | *** |
| *Characteristics of agricultural lands:* | | | | | | | | |
| Total cultivated area (dunum) | 11.543 | 30.980 | 19.698 | 54.040 | 10.640 | 27.101 | 9.058 | *** |
| Owning two or more farmlands (1 = Yes) | 0.567 | 0.496 | 0.600 | 0.490 | 0.563 | 0.496 | 0.037 | *** |
| Availability of irrigation (1 = Yes) | 0.129 | 0.335 | 0.240 | 0.427 | 0.117 | 0.321 | 0.123 | *** |
| *Characteristics of agricultural products:* | | | | | | | | |
| Vegetable producer (1 = Yes) | 0.152 | 0.359 | 0.264 | 0.441 | 0.139 | 0.346 | 0.125 | *** |
| Fruit and olive producer (1 = Yes) | 0.898 | 0.302 | 0.817 | 0.387 | 0.907 | 0.290 | -0.091 | *** |
| Field crop producer (1 = Yes) | 0.209 | 0.406 | 0.301 | 0.459 | 0.198 | 0.399 | 0.103 | *** |
| Mixed farmer with livestock (1 = Yes) | 0.184 | 0.388 | 0.296 | 0.457 | 0.172 | 0.377 | 0.124 | *** |
| *Governorates:* | | | | | | | | |
| • Jenin (1 = Yes) | 0.161 | 0.368 | 0.167 | 0.373 | 0.161 | 0.367 | 0.006 | |
| • Tubas (1 = Yes) | 0.031 | 0.173 | 0.067 | 0.250 | 0.027 | 0.161 | 0.040 | *** |
| • Tulkarm (1 = Yes) | 0.095 | 0.294 | 0.086 | 0.280 | 0.096 | 0.295 | -0.010 | *** |
| • Nablus (1 = Yes) | 0.152 | 0.359 | 0.157 | 0.364 | 0.151 | 0.358 | 0.006 | |
| • Qalqiliya (1 = Yes) | 0.055 | 0.228 | 0.063 | 0.243 | 0.054 | 0.227 | 0.009 | *** |
| • Salfit (1 = Yes) | 0.056 | 0.229 | 0.063 | 0.244 | 0.055 | 0.227 | 0.009 | *** |
| • Ramallah & Al-Bireh (1 = Yes) | 0.127 | 0.332 | 0.054 | 0.226 | 0.135 | 0.341 | -0.081 | *** |
| • Jericho & Al-Aghwar (1 = Yes) | 0.009 | 0.094 | 0.036 | 0.187 | 0.006 | 0.077 | 0.030 | *** |

*(Continued)*

**Table 1.** (Continued)

| Variables | Total holdings (n = 79,446) | | Holdings that received agricultural extension (n = 7,922) | | Holdings that did not receive agricultural extension (n = 71,524) | | The difference in the means | |
|---|---|---|---|---|---|---|---|---|
| | Mean | S.D. | Mean | S.D. | Mean | S.D. | | |
| • Jerusalem (1 = Yes) | 0.025 | 0.157 | 0.039 | 0.193 | 0.024 | 0.152 | 0.015 | *** |
| • Bethlehem (1 = Yes) | 0.074 | 0.262 | 0.079 | 0.269 | 0.074 | 0.262 | 0.005 | |
| • Hebron (1 = Yes) | 0.215 | 0.411 | 0.188 | 0.391 | 0.218 | 0.413 | -0.030 | *** |

(Note)

*, **, *** indicate significant at the 10%, 5%, 1% level, respectively.

significantly influenced the adoption of all technologies: improved crop varieties, chemical fertilizers, organic fertilizers, pesticides, and biological control after controlling for the influence of the covariates.

Since the microdata used in this study is cross-sectional data, the value of adjusted R-squared is not so high. However, the following could be interpreted from the results. In this analysis, distance means remoteness from the Israeli Segregation Wall and Israeli Settlements. As conflict-related covariates, the distance from the Israeli Segregation Wall hindered the adoption of chemical fertilizers and biological controls but encouraged the adoption of organic fertilizers. In other words, agricultural holdings closer to the Israeli Segregation Wall were more likely to adopt chemical fertilizers and biological controls and less likely to adopt organic fertilizers. Agricultural holdings close to the Israeli Segregation Wall tend to have their farmland confiscated for the construction or expansion of the Israeli Segregation Wall, resulting in a decrease in cultivated area. Therefore, to continue farming on their limited farmland, it is presumed that they tend to adopt chemical fertilizers that are effective in improving crop production per area. Longer distance from Israeli settlements is shown to negatively influence the adoption of organic fertilizers and pesticides but positively influenced the adoption of improved crop varieties and chemical fertilizers. In other words, agricultural holdings closer to Israeli settlements were more likely to adopt organic fertilizers and pesticides and less likely to adopt improved crop varieties and chemical fertilizers.

Regarding land classification by the Oslo II Accord, agricultural holdings located in Area B were more likely to adopt improved crop varieties, pesticides, and biological controls than those in Area A. The holdings located in Area C were more likely to adopt improved crop varieties and less likely to adopt chemical fertilizers and pesticides than those in Area A.

### Estimation of propensity scores

In this study, the propensity scores represent determinants to receive agricultural extension for agricultural holdings. The propensity scores were calculated from multiple variables that should influence determinants to receive agricultural extension and other control variables. Those variables selected for the calculation were influence of the conflicts, characteristics of agricultural holdings, lands and agricultural products, and governorates based on previous studies [24, 28, 36]. Following Ali et al. [37] revealing differences in technology adoption by gender, related dummy variables were applied. The theoretical model for the propensity score calculation is shown in Fig 1.

Table 3 shows the results of the estimation by the logit model. The results show that the distance from the Israeli Segregation Wall and Israeli settlements prevented agricultural holdings from receiving agricultural extension. In other words, agricultural holdings closer to the Israeli

**Table 2. Estimated coefficients by OLS regression on the linear probability model.**

| Dependent variable<br>Independent variable and covariate | Improved crop varieties | | Chemical fertilizers | | Organic fertilizers | | Pesticides | | Biological control | |
|---|---|---|---|---|---|---|---|---|---|---|
| Receiving agricultural extension | 0.071 | *** | 0.078 | *** | 0.056 | *** | 0.073 | *** | 0.028 | *** |
| | (0.005) | | (0.005) | | (0.005) | | (0.005) | | (0.005) | |
| *Influence of the conflict*: | | | | | | | | | | |
| Distance from the Israeli Segregation Wall | -0.001 | | -0.001 | *** | 0.004 | *** | -0.000 | | -0.001 | *** |
| | (0.000) | | (0.000) | | (0.000) | | (0.000) | | (0.000) | |
| Distance from Israeli Settlements | 0.005 | *** | 0.007 | *** | -0.010 | *** | -0.004 | *** | 0.001 | |
| | (0.001) | | (0.001) | | (0.001) | | (0.001) | | (0.001) | |
| Land Classification by the Oslo II Accord (Base: Area A): | | | | | | | | | | |
| • Area B | 0.017 | *** | 0.002 | | -0.003 | | 0.012 | *** | 0.007 | *** |
| | (0.003) | | (0.003) | | (0.004) | | (0.004) | | (0.003) | |
| • Area C | 0.015 | *** | -0.036 | *** | -0.005 | | -0.039 | *** | 0.005 | |
| | (0.005) | | (0.006) | | (0.006) | | (0.006) | | (0.005) | |
| • Israeli Declared East Jerusalem | -0.031 | | 0.150 | *** | -0.005 | | 0.134 | *** | 0.028 | |
| | (0.036) | | (0.049) | | (0.050) | | (0.051) | | (0.032) | |
| • Hebron H1 | -0.014 | | 0.178 | *** | 0.009 | | 0.200 | *** | 0.047 | *** |
| | (0.011) | | (0.014) | | (0.014) | | (0.014) | | (0.012) | |
| • Nature Reserve | -0.236 | *** | -0.226 | *** | -0.448 | *** | -0.320 | *** | -0.107 | *** |
| | (0.038) | | (0.021) | | (0.044) | | (0.032) | | (0.027) | |
| *Characteristics of agricultural holdings*: | | | | | | | | | | |
| Age of holder | -0.000 | *** | 0.000 | *** | -0.000 | * | 0.001 | *** | -0.000 | |
| | (0.000) | | (0.000) | | (0.000) | | (0.000) | | (0.000) | |
| Male holder | 0.013 | *** | 0.031 | ** | 0.062 | *** | 0.052 | *** | 0.025 | *** |
| | (0.005) | | (0.006) | | (0.007) | | (0.007) | | (0.004) | |
| Educational background (Base: Preparatory or less): | | | | | | | | | | |
| • Secondary or associate diploma | 0.016 | *** | 0.033 | *** | 0.016 | *** | 0.039 | *** | 0.010 | *** |
| | (0.003) | | (0.004) | | (0.004) | | (0.004) | | (0.003) | |
| • Bachelor or above | 0.010 | ** | 0.049 | *** | 0.026 | *** | 0.037 | *** | 0.001 | |
| | (0.004) | | (0.005) | | (0.005) | | (0.005) | | (0.004) | |
| Main occupation: Agriculture | 0.031 | *** | 0.058 | *** | 0.047 | *** | 0.073 | *** | 0.024 | *** |
| | (0.004) | | (0.004) | | (0.004) | | (0.004) | | (0.004) | |
| The main purpose of production: For sale | 0.057 | *** | 0.155 | *** | 0.063 | *** | 0.149 | *** | 0.051 | *** |
| | (0.004) | | (0.005) | | (0.005) | | (0.005) | | (0.004) | |
| Management by the holder | -0.016 | *** | 0.003 | *** | -0.010 | *** | 0.022 | *** | -0.004 | |
| | (0.003) | | (0.003) | | (0.004) | | (0.004) | | (0.003) | |
| Total members of a holding | -0.000 | | -0.001 | ** | 0.001 | ** | -0.002 | *** | 0.000 | |
| | (0.000) | | (0.001) | | (0.001) | | (0.001) | | (0.000) | |
| *Characteristics of agricultural land*: | | | | | | | | | | |
| Total cultivated area | 0.000 | * | 0.000 | *** | 0.000 | *** | 0.000 | *** | 0.000 | *** |
| | (0.000) | | (0.000) | | (0.000) | | (0.000) | | (0.000) | |
| Owning two or more farmlands | 0.012 | *** | 0.015 | *** | 0.029 | *** | 0.048 | *** | 0.009 | *** |
| | (0.003) | | (0.003) | | (0.004) | | (0.004) | | (0.003) | |
| Availability of irrigation | 0.164 | *** | 0.204 | *** | 0.119 | *** | 0.154 | *** | 0.110 | *** |
| | (0.006) | | (0.006) | | (0.005) | | (0.006) | | (0.005) | |
| *Characteristics of agricultural products*: | | | | | | | | | | |
| Vegetable producer | 0.363 | *** | 0.214 | *** | 0.148 | *** | 0.221 | *** | 0.110 | *** |
| | (0.006) | | (0.005) | | (0.005) | | (0.005) | | (0.005) | |

(*Continued*)

**Table 2.** (Continued)

| Dependent variable<br>Independent variable and covariate | Improved crop varieties | | Chemical fertilizers | | Organic fertilizers | | Pesticides | | Biological control | |
|---|---|---|---|---|---|---|---|---|---|---|
| Fruit and olive producer | -0.041 | *** | -0.026 | *** | 0.117 | *** | 0.089 | *** | 0.001 | |
| | (0.006) | | (0.005) | | (0.006) | | (0.006) | | (0.005) | |
| Field crop producer | 0.169 | *** | 0.076 | *** | 0.042 | *** | 0.060 | *** | 0.015 | *** |
| | (0.004) | | (0.004) | | (0.004) | | (0.005) | | (0.004) | |
| Mixed farmer with livestock | -0.013 | *** | -0.075 | *** | 0.107 | *** | -0.042 | *** | 0.011 | *** |
| | (0.004) | | (0.004) | | (0.004) | | (0.005) | | (0.004) | |
| Constant | 0.115 | *** | 0.117 | *** | 0.431 | *** | 0.214 | *** | 0.036 | *** |
| | (0.012) | | (0.013) | | (0.015) | | (0.014) | | (0.011) | |
| Governorate dummy | Yes | | Yes | | Yes | | Yes | | Yes | |
| Adjusted *R*-squared | 0.282 | | 0.207 | | 0.081 | | 0.191 | | 0.111 | |
| Number of observations | 79,446 | | 79,446 | | 79,446 | | 79,446 | | 79,446 | |

(Note) The figures in parentheses represent robust standard errors.

*, **, *** indicate significant at the 10%, 5%, 1% level, respectively.

Segregation Wall and Israeli settlements were more likely to receive agricultural extension. Regarding other covariates, most of the variables other than the production of fruit and olive trees and the location in Area B have a positive effect on receiving agricultural extension. As mentioned, the Directorates of Agriculture provide agricultural extension services including supplying farming inputs to help farmers who are facing difficulties arising from conflict [11].

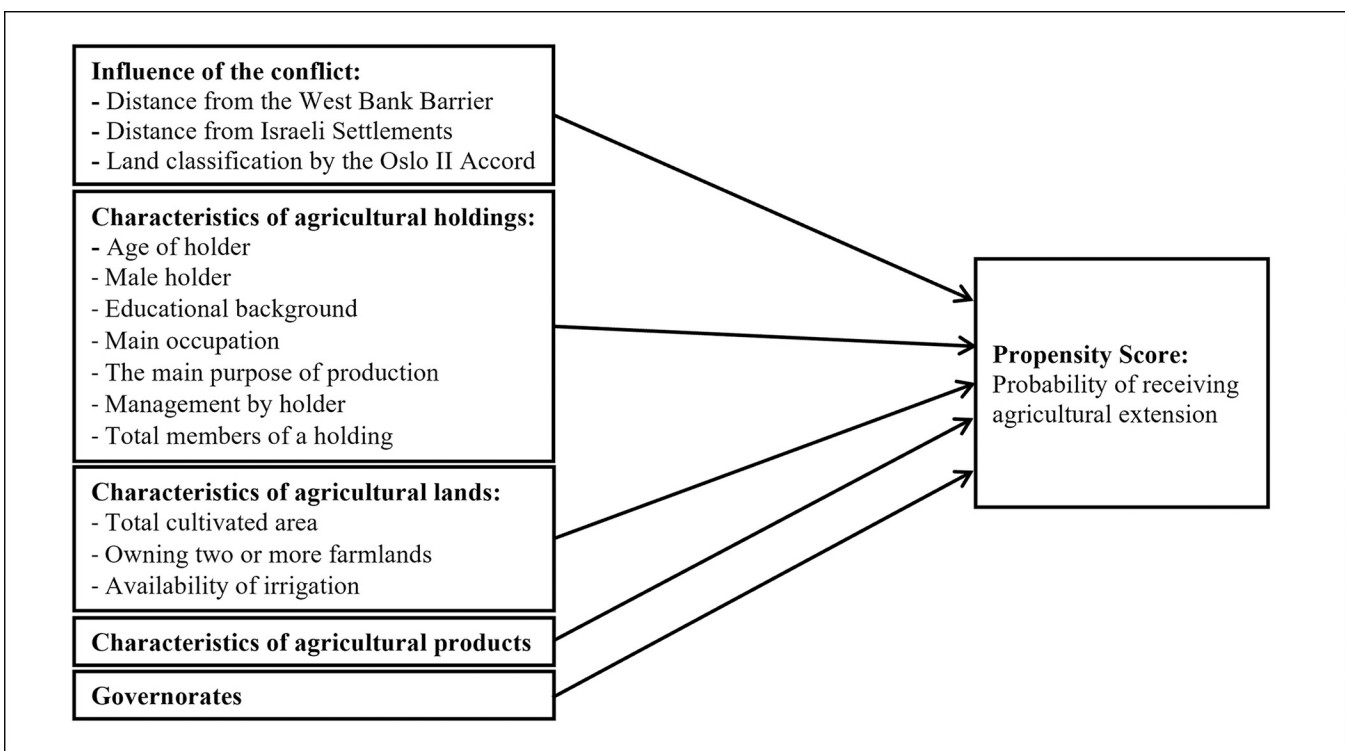

**Fig 1. Theoretical model of propensity score.**

**Table 3. Determinants to receive agricultural extension for agricultural holdings.**

| Covariate | Coef. | | Robust Std. Err. | Average Marginal Effect |
|---|---|---|---|---|
| *Influence of the conflict*: | | | | |
| Distance from the Israeli Segregation Wall | -0.031 | *** | 0.004 | -0.003 |
| Distance from Israeli Settlements | -0.021 | *** | 0.005 | -0.002 |
| Land classification by the Oslo II Accord (Base: Area A): | | | | |
| • Area B | -0.224 | *** | 0.030 | -0.019 |
| • Area C | 0.066 | | 0.042 | 0.006 |
| • Israeli Declared East Jerusalem | 0.381 | | 0.245 | 0.040 |
| • Hebron H1 | 0.288 | *** | 0.107 | 0.029 |
| • Nature Reserve | 1.724 | *** | 0.240 | 0.270 |
| *Characteristics of agricultural holdings*: | | | | |
| Age of holder | 0.001 | | 0.001 | 0.000 |
| Male holder | 0.133 | ** | 0.055 | 0.011 |
| Educational background (Base: Preparatory or less): | | | | |
| • Secondary or associate diploma | 0.062 | ** | 0.031 | 0.005 |
| • Bachelor or above | 0.165 | *** | 0.039 | 0.015 |
| Main occupation: Agriculture | 0.325 | *** | 0.031 | 0.028 |
| The main purpose of production: For sale | 0.363 | *** | 0.033 | 0.031 |
| Management by holder | -0.002 | | 0.028 | 0.000 |
| Total members of a holding | 0.011 | ** | 0.005 | 0.001 |
| *Characteristics of agricultural lands*: | | | | |
| Total cultivated area | 0.002 | *** | 0.000 | 0.000 |
| Owning two or more farmlands | -0.019 | | 0.028 | -0.002 |
| Availability of irrigation | 0.381 | *** | 0.038 | 0.032 |
| *Characteristics of agricultural products*: | | | | |
| Vegetable producer | 0.237 | *** | 0.036 | 0.020 |
| Fruit and olive producer | -0.103 | *** | 0.040 | -0.009 |
| Field crop producer | 0.229 | *** | 0.032 | 0.020 |
| Mixed farmer with livestock | 0.396 | *** | 0.031 | 0.034 |
| Constant | -2.469 | *** | 0.109 | |
| Governorate (dummy) | Yes | | | |
| Pseudo-$R$-squared | 0.064 | | | |
| Number of observations | 79,446 | | | |

(Note)

*, **, *** indicate significant at the 10%, 5%, 1% level, respectively.

Based on these effects, agricultural holdings near the Israeli Segregation Walls and Israeli settlements are more likely to receive agricultural extension. Regarding land classification by the Oslo II Accord, agricultural holdings located in Area B are less likely to receive agricultural extension than those in Area A.

Since the microdata used in this study is cross-sectional data, the value of pseudo-R-squared is not so high. However, the following could be interpreted from the results. Generally, agriculture is a family business in the West Bank, so most female farmers work on family farms. Female farmers work in farming and agro-processing on family farms and contribute significantly to the agricultural sector and GDP in the West Bank [38]. As mentioned, most of the extension officers of the Palestinian Authority as of 2010 were male officers. In general, female farmers prefer female extension officers [39]. But number of female extension officers are

insufficient, thereby hindering outreach to female farmers since cultural traditions often restrict male extension agents from providing services to female farmers. It seems to support the estimation results. Regarding the holders' educational background, the holders with secondary or associate diplomas and bachelor's degrees or above are more likely to receive agricultural extension than those with preparation or less. Several previous studies have shown that the higher education level of farmers positively influences their use of information sources [40–42]. It is presumed that the higher the holder's educational background, the higher the interest in new technologies and the higher the frequency of contacting extension officers. Furthermore, the results would indicate that full-time and commercial farmers are more interested in improving yields and agricultural income by adopting new technologies and farm inputs through agricultural extension. Additionally, the results show that the agricultural holdings with irrigated farmland are more likely to receive agricultural extension. This may indicate that irrigated agricultural lands are more effective in adopting new technologies. Furthermore, as covariates related to the characteristics of agricultural products, vegetable producers, field crop producers, and mixed farmers with livestock are more likely to receive agricultural extension than fruit and olive producers. Fruit trees and olives are permanent crops, and the effect of applying the new technologies is slow to appear as a yield; therefore, it is presumed that these farmers are less interested in acquiring the new technologies.

## Estimation of the average treatment effect on the treated

The second step was matching agricultural holdings between treated and control groups based on calculated propensity scores and estimation of the ATT. In this estimation, the treated group included agricultural holdings that had received agricultural extension, and the control group included holdings that had not received agricultural extension. Single nearest-neighbor, Kernel, and Radius matching methods were employed to confirm the robustness of the PSM estimation. Regarding the single nearest-neighbor matching, the conditions set were matching without a caliper and 1-to-1 matching without replacement. As for Kernel matching, Gaussian Kernel matching with the bandwidth of 0.01 was applied. Austin [43] recommended matching using a caliper of width 0.2 of the standard deviation of the logit of the propensity scores when estimating the mean differences. According to it, for Radius matching, the value (0.01) obtained by multiplying the standard deviation of the propensity scores by 0.2 was set as the caliper for the matching.

Table 4 shows the ATT of agricultural extension on technology adoption. In this study, ATT represents a difference in the average values between the treated and control groups. Regarding technology adoption, a dummy variable with 1 for agricultural holdings that had adopted the technologies and 0 for those that had not were used; therefore, the ATT in this

**Table 4. Results of ATT of agricultural extension on technology adoption.**

| Adoption of technologies (Dependent variables) | Nearest neighbor matching | | Kernel matching (bandwidth = 0.01) | | Radius matching (caliper = 0.01) | |
|---|---|---|---|---|---|---|
| | ATT | S.E. | ATT | S.E. | ATT | S.E. |
| Improved crop varieties | 0.071 *** | 0.008 | 0.076 *** | 0.006 | 0.075 *** | 0.006 |
| Chemical fertilizers | 0.077 *** | 0.008 | 0.080 *** | 0.006 | 0.079 *** | 0.006 |
| Organic fertilizers | 0.054 *** | 0.007 | 0.056 *** | 0.006 | 0.055 *** | 0.006 |
| Pesticides | 0.068 *** | 0.008 | 0.074 *** | 0.006 | 0.073 *** | 0.006 |
| Biological control | 0.038 *** | 0.006 | 0.033 *** | 0.005 | 0.033 *** | 0.005 |

(Note)

*, **, *** indicate significant at the 10%, 5%, 1% level, respectively.

table represents the differences (percentage points) of the technology adoption rates between the treated and control groups.

For example, looking at nearest-neighbor matching, ATT was 0.071, which was a difference between the treated and control groups in the average values, implying that the ATT of agricultural extension on the technology adoption after controlling for the propensity of receiving agricultural extension was 7.1 percentage points. Similarly, for chemical fertilizers, organic fertilizers, pesticides, and biological controls, the estimated ATT was 7.7, 5.4, 6.8, and 3.8 percentage points, respectively. A similar trend was observed in all matching methods adopted in this study. Therefore, the estimation results showed that agricultural extension would have a positive impact on the adoption of all technologies estimated in this study.

## Balancing test of the PSM

To evaluate the matching reliability, a balancing test was conducted to compare the differences in the mean values of the covariates between the treated and control groups before and after the matching. Table 5 shows the results of the balancing test, which compared the differences. Before the matching, significant differences were observed in most of the covariates, except for age, educational background of the holder, management by the holder, and the location of Hebron H1. Hebron H1 in Hebron city is an area administered by the Palestinian Authority under the Hebron Protocol of 1997 [44]. However, significant differences disappeared in all covariates after the matching. Furthermore, the results show that the pseudo-*R*-squared decreased from 0.064 to 0.001 for nearest-neighbor matching and less than 0.001 for Kernel and Radius matchings after the treatment, implying that the logit regression after the matching does not have sufficient explanatory power. Hence, the balancing test confirmed that there are no significant differences among the covariates used for the matching between the treated and control groups after the matching as a new control group. Moreover, the standardized differences (%bias) for the mean values of all covariates between the treated and control groups were 13.2 before the matching and decreased to 1.0 for nearest-neighbor matching, and 0.7 for Kernel and Radius matchings. This indicates that the covariates' influence is successfully mitigated by the matchings, and the ATT could be adequately estimated.

The balance plots of standardized % bias across covariates in nearest-neighbor, Kernel, and Radius matching are, respectively, presented in Figs 2–4 for visual inspection. As expected, standardized % bias across covariations is mitigated in all matching methods.

## Oster's coefficient stability

Following Oster et al. [33], the coefficient stability was tested to verify the robustness of the results. The code "psacalc" of STATA was applied to the estimation of $\delta$ and coefficient bands. The ratio of selection on unobservables to that on observables ($\delta$) captured the required extent of the effect of unobservables in proportion to the effect of observables such that the treatment effect of agricultural extension becomes 0, given a maximum value of *R*-squared. $\delta$ assuming Rmax = 1.3 *R*-squared and $\beta$ = 0 was calculated. The identified value was calculated assuming Rmax = 1.3 *R*-squared with $\delta$ = 1. Table 6 shows the results of the sensitivity test for the selection of unobservables. For example, the value of $\delta$ was 1.922 in the effect of agricultural extension on the adoption of the technologies, indicating that unobservables should be at least 1.922 times larger than observable covariates to drive the treatment effect to zero. All estimates were robust because $\delta$ was larger than 1. Additionally, all identified sets did not include zero, which means that the estimated coefficients were reasonably stable. These results suggested that the estimated coefficients of the treatment effects by agricultural extension were sufficiently robust compared to unobservable heterogeneity.

**Table 5. Results of balancing tests of matching.**

| Covariates | Unmatched | | | | | Nearest-neighbor matching | | | | |
|---|---|---|---|---|---|---|---|---|---|---|
| | Mean | | Diff. | p>\|t\| | % bias | Mean | | Diff. | p>\|t\| | % bias |
| | Treated | Control | | | | Treated | Control | | | |
| *Influence of the conflict:* | | | | | | | | | | |
| Distance from the Israeli Segregation Wall | 5.717 | 5.498 | 0.219 | 0.000 | 4.4 | 5.716 | 5.683 | 0.034 | 0.691 | 0.7 |
| Distance from Israeli Settlements | 4.127 | 3.926 | 0.201 | 0.000 | 6.1 | 4.127 | 4.088 | 0.039 | 0.484 | 1.2 |
| *Land classification by the Oslo II Accord (Base: Area A):* | | | | | | | | | | |
| • Area B | 0.448 | 0.536 | -0.088 | 0.000 | -17.7 | 0.448 | 0.453 | -0.006 | 0.482 | -1.1 |
| • Area C | 0.132 | 0.078 | 0.055 | 0.000 | 17.8 | 0.132 | 0.131 | 0.002 | 0.724 | 0.6 |
| • Israeli Declared East Jerusalem | 0.003 | 0.001 | 0.002 | 0.000 | 3.7 | 0.003 | 0.003 | -0.000 | 0.879 | -0.3 |
| • Hebron H1 | 0.015 | 0.016 | -0.001 | 0.645 | -0.6 | 0.015 | 0.017 | -0.002 | 0.446 | -1.2 |
| • Nature Reserve | 0.005 | 0.001 | 0.004 | 0.000 | 8.1 | 0.005 | 0.004 | 0.001 | 0.546 | 1.2 |
| *Characteristics of agricultural holding:* | | | | | | | | | | |
| Age of holder | 51.647 | 51.853 | -0.206 | 0.213 | -1.5 | 51.647 | 51.621 | 0.026 | 0.905 | 0.2 |
| Male holder | 0.947 | 0.927 | 0.020 | 0.000 | 8.3 | 0.947 | 0.951 | -0.003 | 0.347 | -1.4 |
| *Educational background of the holder (Base: Preparatory or less):* | | | | | | | | | | |
| • Secondary or associate diploma | 0.220 | 0.222 | -0.001 | 0.784 | -0.3 | 0.220 | 0.223 | -0.003 | 0.674 | -0.7 |
| • Bachelor or above | 0.123 | 0.122 | 0.000 | 0.964 | 0.1 | 0.122 | 0.126 | -0.003 | 0.531 | -1.0 |
| Main occupation: Agriculture | 0.357 | 0.200 | 0.157 | 0.000 | 35.6 | 0.357 | 0.352 | 0.005 | 0.517 | 1.1 |
| The main purpose of production: For sale | 0.339 | 0.170 | 0.169 | 0.000 | 39.6 | 0.339 | 0.340 | -0.001 | 0.907 | -0.2 |
| Management by holder | 0.727 | 0.724 | 0.003 | 0.577 | 0.7 | 0.727 | 0.731 | -0.004 | 0.532 | -1.0 |
| Total members of a holding | 6.384 | 6.190 | 0.195 | 0.000 | 6.9 | 6.384 | 6.379 | 0.005 | 0.905 | 0.2 |
| *Characteristics of agricultural land:* | | | | | | | | | | |
| Total cultivated area | 19.698 | 10.640 | 9.058 | 0.000 | 21.2 | 19.401 | 18.876 | 0.525 | 0.530 | 1.2 |
| Owning two or more farmlands | 0.600 | 0.563 | 0.037 | 0.000 | 7.5 | 0.600 | 0.600 | -0.000 | 0.987 | 0.0 |
| Availability of irrigation | 0.240 | 0.117 | 0.123 | 0.000 | 32.5 | 0.239 | 0.236 | 0.004 | 0.588 | 1.0 |
| *Characteristics of agricultural products:* | | | | | | | | | | |
| Vegetable producer | 0.264 | 0.139 | 0.125 | 0.000 | 31.5 | 0.264 | 0.267 | -0.002 | 0.746 | -0.6 |
| Fruit and olive producer | 0.817 | 0.907 | -0.091 | 0.000 | -26.5 | 0.817 | 0.822 | -0.005 | 0.397 | -1.5 |
| Field crop producer | 0.301 | 0.198 | 0.103 | 0.000 | 24.0 | 0.301 | 0.302 | -0.001 | 0.931 | -0.1 |
| Mixed farmer with livestock | 0.296 | 0.172 | 0.124 | 0.000 | 29.6 | 0.296 | 0.295 | 0.001 | 0.889 | 0.2 |
| *Governorates (Base: Jenin):* | | | | | | | | | | |
| • Tubas | 0.067 | 0.027 | 0.040 | 0.000 | 19.2 | 0.067 | 0.063 | 0.004 | 0.287 | 2.0 |
| • Tulkarm | 0.086 | 0.096 | -0.010 | 0.003 | -3.6 | 0.086 | 0.084 | 0.002 | 0.608 | 0.8 |
| • Nablus | 0.157 | 0.151 | 0.007 | 0.126 | 1.8 | 0.157 | 0.162 | -0.005 | 0.386 | -1.4 |
| • Qalqiliya | 0.063 | 0.054 | 0.009 | 0.001 | 3.8 | 0.063 | 0.066 | -0.003 | 0.458 | -1.2 |
| • Salfit | 0.063 | 0.055 | 0.009 | 0.001 | 3.7 | 0.064 | 0.070 | -0.006 | 0.118 | -2.6 |
| • Ramallah & Al-Bireh | 0.054 | 0.135 | -0.081 | 0.000 | -27.8 | 0.054 | 0.053 | 0.001 | 0.724 | 0.4 |
| • Jericho & Al-Aghwar | 0.036 | 0.006 | 0.030 | 0.000 | 21.2 | 0.036 | 0.036 | 0.000 | 1.000 | 0.0 |
| • Jerusalem | 0.039 | 0.024 | 0.015 | 0.000 | 8.6 | 0.039 | 0.041 | -0.002 | 0.441 | -1.4 |
| • Bethlehem | 0.079 | 0.074 | 0.005 | 0.134 | 1.8 | 0.079 | 0.082 | -0.004 | 0.365 | -1.5 |
| • Hebron | 0.188 | 0.218 | -0.030 | 0.000 | -7.4 | 0.188 | 0.179 | 0.010 | 0.109 | 2.4 |
| Mean bias of covariates | | | | | 13.2 | | | | | 1.0 |
| Pseudo-*R*-squared | 0.064 | | | | | 0.001 | | | | |

| Covariates | Kernel matching (bandwidth = 0.01) | | | | | Radius matching (caliper = 0.01) | | | | |
|---|---|---|---|---|---|---|---|---|---|---|
| | Mean | | Diff. | p>\|t\| | % bias | Mean | | Diff. | p>\|t\| | % bias |
| | Treated | Control | | | | Treated | Control | | | |
| *Influence of the conflict:* | | | | | | | | | | |

*(Continued)*

**Table 5.** (Continued)

| | | | | | | | | | | |
|---|---|---|---|---|---|---|---|---|---|---|
| Distance from the Israeli Segregation Wall | 5.716 | 5.706 | 0.011 | 0.900 | 0.2 | 5.715 | 5.700 | 0.016 | 0.853 | 0.3 |
| Distance from Israeli Settlements | 4.127 | 4.129 | -0.002 | 0.967 | -0.1 | 4.127 | 4.129 | -0.002 | 0.972 | -0.1 |
| *Land classification by the Oslo II Accord (Base: Area A):* | | | | | | | | | | |
| • Area B | 0.448 | 0.448 | -0.000 | 0.990 | 0.0 | 0.448 | 0.447 | 0.001 | 0.935 | 0.1 |
| • Area C | 0.132 | 0.128 | 0.004 | 0.429 | 1.4 | 0.132 | 0.129 | 0.004 | 0.499 | 1.2 |
| • Israeli Declared East Jerusalem | 0.003 | 0.003 | 0.000 | 0.880 | 0.3 | 0.003 | 0.003 | 0.000 | 0.893 | 0.3 |
| • Hebron H1 | 0.015 | 0.016 | -0.001 | 0.757 | -0.5 | 0.015 | 0.016 | -0.001 | 0.776 | -0.5 |
| • Nature Reserve | 0.005 | 0.004 | 0.001 | 0.561 | 1.2 | 0.005 | 0.004 | 0.001 | 0.623 | 1.0 |
| *Characteristics of agricultural holding:* | | | | | | | | | | |
| Age of holder | 51.647 | 51.692 | -0.045 | 0.839 | -0.3 | 51.645 | 51.705 | -0.060 | 0.786 | -0.4 |
| Male holder | 0.947 | 0.948 | -0.000 | 0.982 | 0.0 | 0.947 | 0.948 | -0.000 | 0.947 | -0.1 |
| *Educational background of the holder (Base: Preparatory or less):* | | | | | | | | | | |
| • Secondary or associate diploma | 0.220 | 0.222 | -0.002 | 0.806 | -0.4 | 0.220 | 0.222 | -0.002 | 0.775 | -0.5 |
| • Bachelor or above | 0.122 | 0.123 | -0.001 | 0.892 | -0.2 | 0.122 | 0.123 | -0.001 | 0.882 | -0.2 |
| Main occupation: Agriculture | 0.357 | 0.353 | 0.004 | 0.607 | 0.9 | 0.357 | 0.354 | 0.002 | 0.759 | 0.5 |
| The main purpose of production: For sale | 0.339 | 0.335 | 0.004 | 0.624 | 0.9 | 0.339 | 0.337 | 0.002 | 0.812 | 0.4 |
| Management by holder | 0.727 | 0.731 | -0.004 | 0.559 | -0.9 | 0.727 | 0.731 | -0.004 | 0.600 | -0.8 |
| Total members of a holding | 6.384 | 6.378 | 0.006 | 0.897 | 0.2 | 6.384 | 6.379 | 0.005 | 0.914 | 0.2 |
| *Characteristics of agricultural land:* | | | | | | | | | | |
| Total cultivated area | 19.401 | 18.404 | 0.997 | 0.222 | 2.3 | 19.077 | 18.019 | 1.058 | 0.157 | 2.5 |
| Owning two or more farmlands | 0.600 | 0.603 | -0.002 | 0.763 | -0.5 | 0.600 | 0.603 | -0.003 | 0.688 | -0.6 |
| Availability of irrigation | 0.239 | 0.232 | 0.007 | 0.280 | 1.9 | 0.239 | 0.233 | 0.006 | 0.364 | 1.6 |
| *Characteristics of agricultural products:* | | | | | | | | | | |
| Vegetable producer | 0.264 | 0.258 | 0.006 | 0.354 | 1.6 | 0.264 | 0.259 | 0.005 | 0.458 | 1.3 |
| Fruit and olive producer | 0.817 | 0.820 | -0.004 | 0.538 | -1.1 | 0.817 | 0.820 | -0.003 | 0.629 | -0.9 |
| Field crop producer | 0.301 | 0.301 | 0.000 | 0.958 | 0.1 | 0.301 | 0.303 | -0.001 | 0.855 | -0.3 |
| Mixed farmer with livestock | 0.296 | 0.295 | 0.001 | 0.866 | 0.3 | 0.296 | 0.297 | -0.001 | 0.920 | -0.2 |
| *Governorates (Base: Jenin):* | | | | | | | | | | |
| • Tubas | 0.067 | 0.064 | 0.004 | 0.370 | 1.7 | 0.067 | 0.064 | 0.003 | 0.445 | 1.4 |
| • Tulkarm | 0.086 | 0.087 | -0.001 | 0.831 | -0.3 | 0.086 | 0.086 | -0.000 | 0.942 | -0.1 |
| • Nablus | 0.157 | 0.161 | -0.003 | 0.573 | -0.9 | 0.158 | 0.161 | -0.004 | 0.542 | -1.0 |
| • Qalqiliya | 0.063 | 0.064 | -0.001 | 0.885 | -0.2 | 0.063 | 0.064 | -0.001 | 0.886 | -0.2 |
| • Salfit | 0.064 | 0.065 | -0.002 | 0.624 | -0.8 | 0.064 | 0.066 | -0.002 | 0.560 | -1.0 |
| • Ramallah & Al-Bireh | 0.054 | 0.049 | 0.005 | 0.179 | 1.6 | 0.054 | 0.050 | 0.005 | 0.200 | 1.6 |
| • Jericho & Al-Aghwar | 0.036 | 0.034 | 0.002 | 0.443 | 1.6 | 0.036 | 0.034 | 0.002 | 0.429 | 1.6 |
| • Jerusalem | 0.039 | 0.038 | 0.000 | 0.917 | 0.2 | 0.039 | 0.039 | 0.000 | 0.998 | 0.0 |
| • Bethlehem | 0.079 | 0.081 | -0.002 | 0.645 | -0.7 | 0.079 | 0.081 | -0.002 | 0.621 | -0.8 |
| • Hebron | 0.188 | 0.188 | 0.000 | 0.974 | 0.1 | 0.188 | 0.187 | 0.001 | 0.847 | 0.3 |
| Mean bias of covariates | | | | | 0.7 | | | | | 0.7 |
| Pseudo-*R*-squared | | 0.000 | | | | | 0.000 | | | |

## Section 6: Discussion

### Improved crop varieties

Improved crop varieties suitable for local agro-climatic environments and their extension are considered the most important means to enhance crop yield and improve the livelihood of farmers in developing countries [45]. Takahashi et al. [46] summarized past studies on the impact of the adoption of agricultural technology on farmers and showed that improved crop

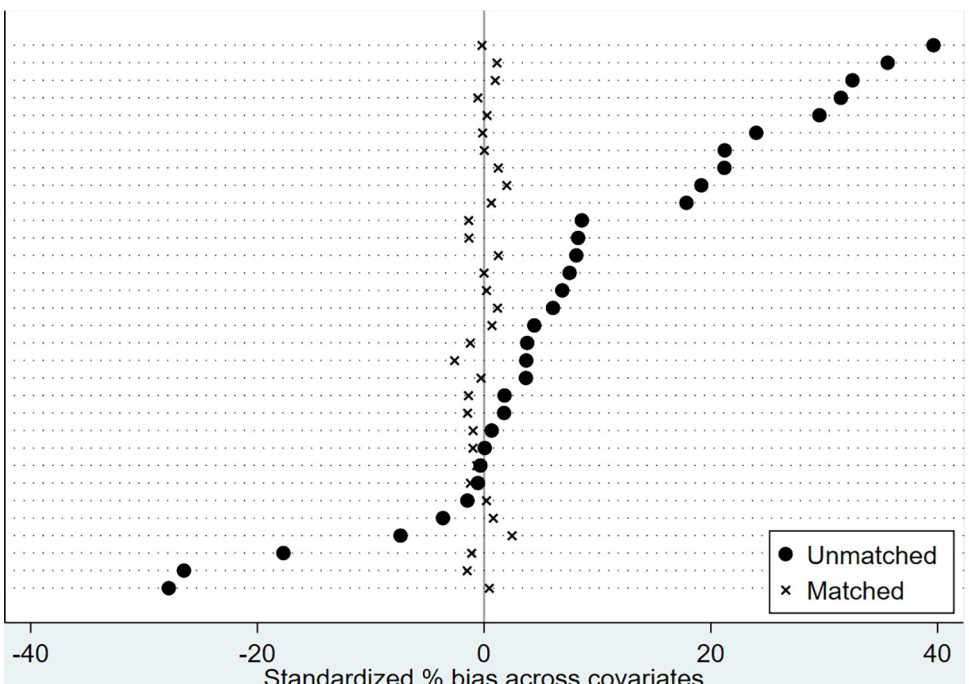

**Fig 2. Balance plots of standardized % bias across covariates (Nearest-neighbor matching).**

varieties generally have positive effects on yield, household income, consumption, poverty reduction, and farmers' welfare.

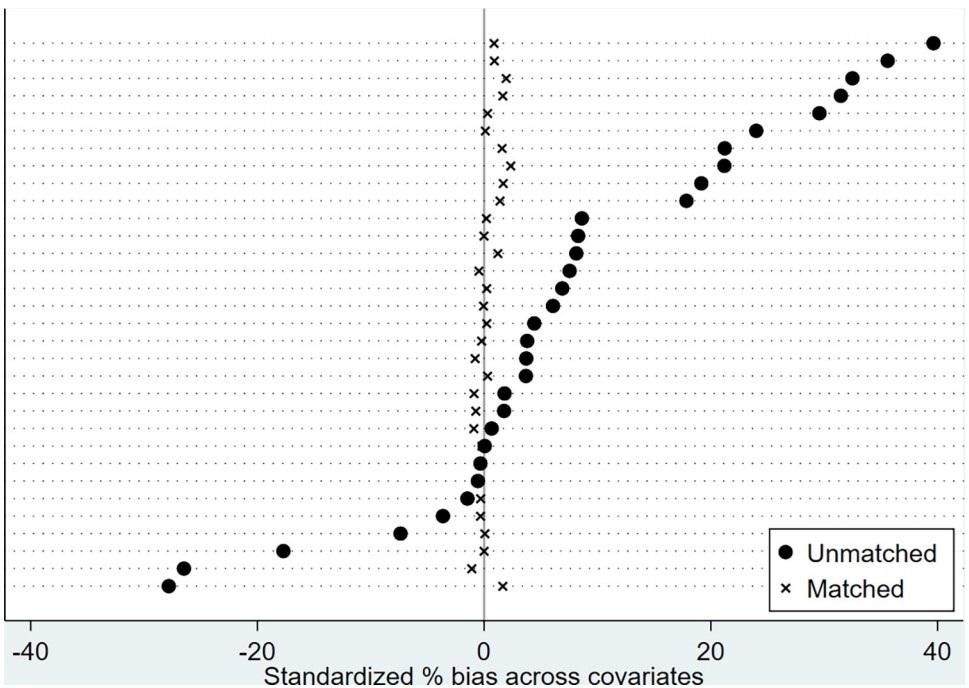

**Fig 3. Balance plots of standardized % bias across covariates (Kernel matching, bandwidth = 0.01).**

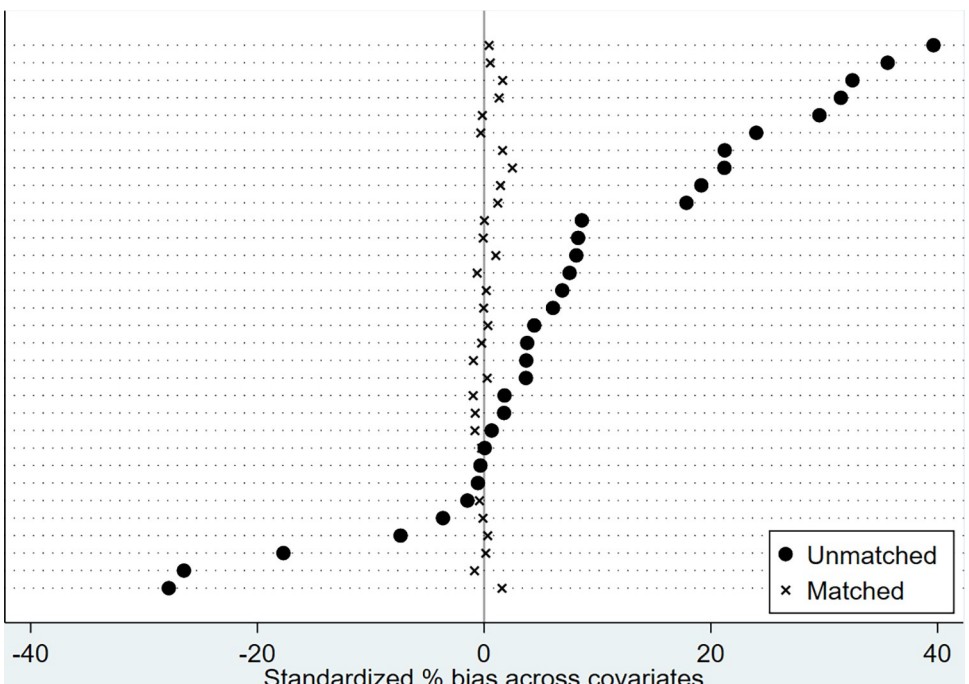

**Fig 4. Balance plots of standardized % bias across covariates (Radius matching, caliper = 0.01).**

Numerous crop varieties that could be adapted to arid regions have been developed. These varieties are also available to farmers in the West Bank. The conflicted-affected agricultural holdings must adopt those varieties with high yields and resistance to pests and diseases to maintain their livelihoods on the limited farmlands. Additionally, as one of the measures of adaptation against climate change, the use of improved drought-resistant varieties could improve agricultural incomes [47]. The best approach to pest management is the use of disease and pest-resistant varieties or breeds in the West Bank. The Palestinian Authority can financially contribute in order to help reduce the impacts of pests on drought-affected farmers [12].

This study proved that agricultural extension promotes the adoption of improved crop varieties in the West Bank. Therefore, it is suggested that the agricultural extension could contribute to yield, household income, consumption, and poverty reduction of agricultural holdings in the West Bank. Furthermore, Yamano et al. [48] stressed the importance of educating farmers about the benefits of stress-tolerant crop varieties through extension activities, and this suggestion is also important for future agricultural extension in the West Bank.

**Table 6. Results of Oster's coefficient stability.**

| Dependent variable | Uncontrolled | | Controlled | | $\delta$ for $\beta = 0$ | Identified set | Identified set includes zero |
|---|---|---|---|---|---|---|---|
| | Coef. | *R*-squared | Coef. | *R*-squared | | | |
| Improved crop varieties | 0.179 | 0.017 | 0.071 | 0.282 | 1.922 | [0.035, 0.071] | No |
| Chemical fertilizers | 0.178 | 0.015 | 0.078 | 0.207 | 2.283 | [0.045, 0.078] | No |
| Organic fertilizers | 0.124 | 0.006 | 0.056 | 0.081 | 2.339 | [0.033, 0.056] | No |
| Pesticides | 0.171 | 0.011 | 0.073 | 0.191 | 2.170 | [0.040, 0.073] | No |
| Biological control | 0.082 | 0.004 | 0.028 | 0.111 | 1.552 | [0.010, 0.028] | No |

## Chemical fertilizers

The adoption of chemical fertilizer is necessary to get the full yield potential of crops. According to the study by Takahashi et al. [46] on the adoption of chemical fertilizers by farmers, a well-known limiting factor was the risks faced by farmers. Based on data in India, Dercon and Christiaensen [49] demonstrated that chemical fertilizer application decreased when a farmer faced downside risk in consumption.

It is estimated that the amount of chemical fertilizer applied per area in the Palestinian territories was approximately 21% of that in Israel and approximately 40% of that in Jordan in 2010 [50]. The price of fertilizers in the occupied Palestinian territories was 120–150 NIS (New Israel Shekel) in 2013 for a 25-kg bag for NPK (nitrogen (N), phosphorus (P) and potassium (K)) 13:13:13 compound fertilizers at a lower concentration [51]. However, the price of a 25-kg bag of NPK 20:20:20 compound fertilizer at standard concentration was NIS75 in Israel [52]. Therefore, chemical fertilizers in the Palestinian territories were priced 1.6 to 2.0 times higher than those in Israel. Since 2008, the Israeli Occupation Authorities have restricted the import of items that could be diverted for military purposes (dual-use goods) into the West Bank and Gaza Strip [51]. The regulatory list also includes the following chemical fertilizers used extensively worldwide [51]: ammonium nitrate, potassium nitrate, urea, urea nitrate, NPK 17-10-27 compound fertilizer, NPK 20-20-20 compound fertilizer, etc. Therefore, few varieties of chemical fertilizers are available in the Palestinian territories, which limits farmers' options.

Checkpoints on the West Bank–Israel and West Bank–Jordan borders have transportation restrictions, resulting in frequent delays and blockages [6]. Checkpoints within the West Bank are also frequently closed owing to the security situation, thus slowing the movement of agricultural products within the West Bank [6]. These movement restrictions also affect the market prices of agricultural products, thus destabilizing farm management. These risks may be psychologically preventing agricultural holdings from using chemical fertilizers.

## Organic fertilizers

Soil fertility must be maintained for sustainable agriculture. Organic fertilizers should be applied to make the physical, chemical, and biological conditions of the soil suitable for growing crops. Some Palestinian farmers use compost as organic fertilizers made from livestock manure and wheat straw. However, it takes time to derive the benefits of compost. On the other hand, most Palestinian farmers rely on chemical fertilizers as they are effective in increasing crop yields in the short term. The West Bank faces the risk of Palestinian agricultural land being confiscated by the Israeli Occupation Authorities. This may hinder the use of compost as deriving its benefits is time-consuming. This study suggested that mixed farming with animal husbandry positively influences the adoption of organic fertilizers. In the West Bank, some agricultural holdings produce only crops, while some only raise livestock. This suggests that it is important for future agricultural extension to promote collaboration among those farmers so that organic materials derived from livestock farming can be easily used by crop farmers.

## Pesticides

In the West Bank, irrigated and intensive farming particularly depend on pesticides as a major tool for pest and disease control in crop production. Studies have shown that fertilizers constituted 21% of the total cost of inputs, followed by 20% pesticides [52]. Most of these pesticides were purchased in Israel and distributed to Palestinian farmers through merchants and pesticide distributors in the West Bank market [53]. It is known that several pesticides that are

internationally banned or have expired are imported to the West Bank [54] and nearly 50% of the pesticides used in the Palestinian territories were internationally illegal [55]. Furthermore, the labels of pesticides imported into the West Bank do not contain chemical specifications or safety instructions written in Arabic. Most Palestinian farmers cannot understand and use it safely [54]. Therefore, it was suggested that the Palestinian Authority makes proper use of technology to distribute legal pesticides instead of simply disseminating the use of pesticides, which could lead to the safety of crops, maintenance of farmers' health, enhancing consumer confidence, and sustainable agriculture in the West Bank.

### Biological control

As defined in the Agricultural Census 2010, the term biological control means integrated pest and disease management (IPM). It means an integrated control method with less impact on crops, humans, and the environment, which comprehensively combines physical (burning and soil solarization), chemical (chemical pesticides), and biological control (parasites and predators). Knowler et al. [27] summarized past studies on the adoption of conservation agricultural technologies by farmers and found that many studies reported that higher educational background, young age, larger farm size, lower off-farm income, extensive agricultural experience, and receiving agricultural extension are effective in promoting the adoption of conservation agricultural technologies. These results are consistent with the results of this study. In the estimation of the ATT using the PSM method, the number of agricultural holdings adopting biological control was the least compared with other technologies. It seems that conservation agriculture is relatively new in the West Bank, thus revealing the lack of progress in its adoption. Moreover, the West Bank faces the risk of farmland confiscation by the Israeli Occupation Authorities. Therefore, it is speculated that farmers are less interested in the sustainable use of farmland through conservation agriculture.

## Section 7: Conclusions

In this study, the PSM method was used to estimate the effect of agricultural extension on technology adoption by conflict-affected Palestinian farmers. First, it was proved that agricultural extension promotes technology adoption by agricultural holdings using the OLS method for preliminary estimation. Second, using the PSM method, the ATT of agricultural extension was estimated after mitigating the endogenous bias arising from the self-selection of agricultural holdings in choosing to receive agricultural extension. In this study, the distance from the Israeli Segregation Wall and the Israeli settlements and land classification based on the Oslo II Accord was used as covariates for the effects of the conflict. However, the effects of the conflict are diverse and devastating on Palestinian farmers in the OPT (the West Bank and Gaza Strip). They cannot be fully explained by the variables used in this study. Therefore, it is still necessary to find and use variables that more accurately represent the impact of the conflict.

In the West Bank, the Israeli Occupation Authorities continue to confiscate Palestinian farmlands, and many farmers' farmlands have decreased but they must continue their livelihood on their limited farmlands. Therefore, the Palestinian Authority should disseminate effective and appropriate farming methods such as improved crop varieties, chemical fertilizers, and pesticides to increase yields per area. Palestine agricultural holdings also constantly face the risk of losing their lands. In the West Bank, the risk of future farmland loss hinders the adoption of agricultural inputs, particularly organic fertilizers, and biological controls including IPM, that are effective for sustainable agriculture.

This study proved that agricultural extension promotes the adoption of these technologies. Therefore, Palestinian Authority may be able to promote broader adoption of these

technologies by combining methods to increase yield per area with any measure to alleviate farmers' psychological anxieties considering their behavior. Then, it is hoped that future agricultural extension by the Palestinian Authority may result in sustainable agricultural land use and farm management.

The Gaza Strip could not be included in the same model in this study because it has no Israeli settlements. In the future, it will be necessary to make similar estimates and compare them with those in the West Bank regarding the impact of the conflict and the effects of agricultural extension, taking into account the political situation and natural environment in the Gaza Strip. At the time this study was conducted, only individual data from the Agricultural Census 2010 were publicly available. However, in the future, when the Agricultural Census 2020 microdata becomes public and available, updated estimates will be made using it.

## Acknowledgments

The authors would like to thank two anonymous reviewers who provided invaluable feedback on the structure and content of this manuscript, and the PCBS for providing microdata of the Agricultural Census 2010.

## Author Contributions

**Conceptualization:** Nakamura Tomoki.

**Data curation:** Nakamura Tomoki.

**Formal analysis:** Nakamura Tomoki.

**Funding acquisition:** Nakamura Tomoki.

**Investigation:** Nakamura Tomoki.

**Methodology:** Nakamura Tomoki, Kashiwagi Kenichi.

**Project administration:** Nakamura Tomoki, Kashiwagi Kenichi.

**Resources:** Nakamura Tomoki.

**Software:** Nakamura Tomoki, Kashiwagi Kenichi.

**Supervision:** Kashiwagi Kenichi, Ujiie Kiyokazu.

**Validation:** Kashiwagi Kenichi, Ujiie Kiyokazu.

**Visualization:** Nakamura Tomoki.

**Writing – original draft:** Nakamura Tomoki.

**Writing – review & editing:** Nakamura Tomoki.

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
