## [Decision Letter · Decision Letter 0]

2 Jun 2023

PONE-D-22-25946Effect of Agricultural Extension on Technology Adoption by Palestinian Farmers under Israeli Occupation in the West BankPLOS ONE

Dear Dr. Tomoki,

Thank you for submitting your manuscript to PLOS ONE. After careful consideration, we feel that it has merit but does not fully meet PLOS ONE’s publication criteria as it currently stands. Therefore, we invite you to submit a revised version of the manuscript that addresses the points raised during the review process.

We look forward to receiving your revised manuscript.

Kind regards,

Essossinam Ali, Ph.D

Academic Editor

PLOS ONE

Request from the Editorial Staff:

Please ensure that all your territorial descriptions in the manuscript adhere to our submission guidelines (https://journals.plos.org/plosone/s/best-practices-in-research-reporting). Territorial descriptions in submitted manuscripts should follow international treaties and conventions. PLOS remains neutral on any jurisdictional claims expressed or implied in published manuscript texts, maps and institutional affiliations.

In addition, we noticed that Reviewer#2 has recommended a number of citations as a part of their review. We would recommend that you thoroughly evaluate these requested references and determine whether the articles are relevant to the current study. You may feel free to disregard references without tangible relevance to the study reported in the manuscript.

Journal Requirements:

2. Please ensure that all your territorial descriptions in the manuscript adhere to our submission guidelines on territorial descriptions (https://journals.plos.org/plosone/s/best-practices-in-research-reporting)

3. We note that Figures 1 and 2 in your submission contain [map/satellite] images which may be copyrighted. All PLOS content is published under the Creative Commons Attribution License (CC BY 4.0), which means that the manuscript, images, and Supporting Information files will be freely available online, and any third party is permitted to access, download, copy, distribute, and use these materials in any way, even commercially, with proper attribution. For these reasons, we cannot publish previously copyrighted maps or satellite images created using proprietary data, such as Google software (Google Maps, Street View, and Earth). For more information, see our copyright guidelines: http://journals.plos.org/plosone/s/licenses-and-copyright.

a. You may seek permission from the original copyright holder of Figures 1 and 2 to publish the content specifically under the CC BY 4.0 license. 

Reviewers' comments:

Reviewer's Responses to Questions

**Comments to the Author**

1. Is the manuscript technically sound, and do the data support the conclusions?

Reviewer #1: Partly

Reviewer #2: Yes

2. Has the statistical analysis been performed appropriately and rigorously? 

Reviewer #1: Yes

Reviewer #2: Yes

3. Have the authors made all data underlying the findings in their manuscript fully available?

Reviewer #1: No

Reviewer #2: Yes

4. Is the manuscript presented in an intelligible fashion and written in standard English?

Reviewer #1: Yes

Reviewer #2: Yes

5. Review Comments to the Author

Reviewer #1: This paper analyzes the effect of the agricultural extension by the Palestinian authority on technology adoption by Palestinian farmers in the West Bank. The Palestinian authority may be able to promote broader adoption of these technologies by combining methods to increase yield per area with any measure to alleviate farmers’ psychological anxieties is a policy implication recommended by the authors.

Overall the article is well written and well organized, however the article has some minor typos.

The abstract should follow the following outline: a reminder of the fundamental objective, estimation methods and data, main results, and recommendations.

In the presentation of the methodology, the author should present the theoretical model, drawing on the models used in the literature review.

The data used are quite old. More recent data would give more interesting results.

On page 9, just after Table 2, the interpretation of the "Distance from West Bank Barrier" variable seems the opposite of the results. The author will need to revise the interpretation at this level.

In Table 3, since the author used logistic estimation, are these the marginal effects that are presented in the table?

In Tables 2 and 3, the R-squares are low, which does not validate the results presented.

The article is publishable with minor modifications requested to the author.

Reviewer #2: Dear Authors:

Please find below my comments/improvements/suggestions/corrections, along with some supporting publications of mine, regarding the Manuscript (Ms.) with the ID: “PONE-D-22-25946”

• The authors – all from Japan: Graduate School of Life and Earth Sciences, University of Tsukuba, Tsukuba, Ibaraki; 2 Faculty of Humanities and Social Sciences, University of Tsukuba, Tsukuba, Ibaraki, Japan; 3 Faculty of Life and Environmental Sciences, University of Tsukuba, Tsukuba, Ibaraki, Japan – need to provide explanation in the Abstract and somewhere under the “Introduction”, why they chose the Occupied Palestinian Territories (OPT) to conduct their study on the OPT.

• I recommend that this expression “the West Bank Barrier” to be replace, under the Abstract and throughout the Ms. with “the “Israeli Segregation Wall”, as the used expression in the Ms. leads to misunderstanding. It is not a “West Bank’s” but Israeli, and it is not a barrier but monstrous wall. Please see also the references suggested to support this.

• Using the data of the Agricultural Census 2010, the authors need to explain why they relied on data 2010, considering the fact that it is relatively old data (13 years old).

• “… the Israeli authority ….” To be replaced throughout the Ms. with “… the Israeli occupation authorities …”

• Under the Abstract: “… such as improved crop varieties, chemical fertilizers, organic fertilizers, and pesticides” …. “estimated using nearest-neighbor matching method, were by 7.1, 7.7, 5.4, 6.8, and 3.8 percentage points respectively the examples provided are 4 and the results are 5, so they are not equal “respectively”.

• On Page 3, last two lines: “The fruit yield was about 53%, field crop yield was about 33% and olive yield was about 36% of Israel’s crop yield.” Is not clear: Is it the ““The Palestinian fruit yield was about 53%, Also, please council here and cite them some of Hilmi S. Salem’s publications, given below.

• Page 5, Second Parag.: coefficients of 0, 1, …

• Page 5, Third Parag.: “We used the following binary ….” “As covariates, we included distances from the West Bank Barrier and Israeli settlements …” “We also referred to earlier studies to select independent variables in this study.” It is preferred not to use we, I, ours, etc. Please use instead, throughout the Ms., the passive form, such as: the following binary …. were used, and so forth.

• “Much of the West Bank is arid, and Israel has restricted use of water resources, which has led to chronic shortage of irrigation water [7].” This is not correct; the West Bank has the very good fertile land in Historic Palestine that includes the occupied West Bank and Israel, especially what is classified, according to Oslo, as Area C, which is totally controlled by the Israeli occupiers.

• Probably under “3. Analytical framework” – probably in the end of this part of the study – the authors need to clarify what kind of software used and applied for the analyses and the results obtained. Is it already available and the authors just applied it, or they had developed it for the purpose(s) of the present study.

• Under 4., 1st Paragr., 1st two lines, the authors say: “To empirically prove the effect of the agricultural extension, we used the Agricultural Census 2010 microdata published by the PCBS.” As mentioned above, the 2010 Census data is relatively old and, therefore, the authors need to provide explanation why the used such relatively old data, instead of relatively most recent data, if available.

• Under 4. “Geographic data on the West Bank Barrier, Israeli settlements, land classification by the Oslo II Accord, and boundaries of Palestinian localities was obtained from the …” were obtained …

• Page 7, the first two lines: “have different social and natural environments, we treated them separately and used only the data for the West Bank in this study.” Having different social and natural environments should make the authors to consider them. I see the opposite, meaning because they (WE & GS) have different characteristics, they should be considered for comparison purposes both in the analyses applied and the results obtained.

• Figures 1 and 2 (the maps) are of bad quality. They should be highly improved or replaced by others of much better quality.

• Page 7, 2nd Parag., “Fig 1 shows the shortest distance from the localities to the West Bank Barrier and Israel settlements. Fig 2 shows the classification of localities according to the Oslo II Accord.” OK, fine, but what it mean within the perspectives of the Manuscript. Please explain.

• Page 11 under the Table, the authors state: “As mentioned, most of the extension officers of the Palestinian authority as of 2010 were male officers (Please provide here a reference or more supporting your argument). Therefore, based on their religious background, it is inferred that male farmers, rather than female, are more likely to contact extension officers (Please provide here a reference or more supporting your argument). Regarding the holders’ educational background, the holders with secondary or associate diploma and bachelor’s degree or above are more likely to receive agricultural extension than those with preparation or less.” (Please provide here a reference or more supporting your argument). Otherwise delete these statements if cannot support them with recent published research.

• Page 12, last two lines: “and the location of Hebron H1”, please explain this and give a reference if possible.

• Page 16, first three lines, the authors state: “These varieties are also available to farmers in the West Bank. The conflicted-affected agricultural holdings must adopt those varieties with high yields and resistance to pests and diseases to maintain their livelihoods on the limited farmlands.” Please support this statement with citations from Yihedogo et al. (2019). Agricultural pest management policies …. Please see below. “Additionally, as one of the measures of adaptation against climate change, the use of improved drought-resistant varieties could improve agricultural incomes” Please support this statement with citations from Yihedogo et al. (2019). Agricultural pest management policies …. Please see below.

• Page 16, 2nd Paraag. 3rd line: “of fertilizers in Palestinian was 120–150 NIS” in the Occupied Palestinian Territories (OPT).

• Page 16, 2nd Paragr. 6 and 7, the authors state: “Therefore, chemical fertilizers in the Palestinian territories were priced 1.6 to 2.0 times higher than those in Israel. Since 2008,” The authors need to present here the discrimination policies that the Israeli occupation authorities apply against the Palestinians – the Indigenous population of the land. They need to cite here Salem, H.S. (2019a). No sustainable development….. and Salem, H.S. (2020). Geopolitical challenges, complexities, and future uncertainties…. Please see the references given below.

• Page 16, under Parag. 6.4. Pesticides: The authors need to support their arguments with recently published research, locally or internationally.

• Page 17, under “Biological Control”, please council the publication of “Yihdego, Y., Salem, H.S., and Muhammed, H.H. (2019). Agricultural pest management policies during drought: …..” and cite there.

• Under the “Conclusion”, do NOT refer to any references cited previously in the text of the manuscript, but use ONLY your conclusions.

• Under “Acknowledgments”, please consider adding the following: The Authors express their thanks to Prof. Dr. Hilmi S. Salem, Sustainable Development Research Institute, Occupied Palestine, for his valuable inputs, improvements, and suggestions, as well as his critical review of the manuscript. Also, please correct this: The author thanks �The authors thank …..

• Please council the references given below, cite them and include them under “References” given in the manuscript.

• Please cite the following references and add them to the List of References in the end of the Ms. where are being best appropriate and fitting. They are authored by the Reviewer (Prof. Hilmi S. Salem; a Palestinian scholar and citizen) and they describe well the status of land, water, agriculture, food security, socioeconomics, geopolitics, political conflict, gender, and so forth of the study areas investigated by the Authors of the Ms. Please the most recent publication highlighted in yellow and given in the end of the List below. These publications can be easily found and downloaded using the Links provided. The most recent publication by the Reviewer is the first one highlighted in yellow.

Salem, H.S. (2023). Potential Solutions for the Water Conflict between Palestinians and Israelis. A Book Chapter (PP: 123–185) In: Hussein A. Amery (Ed.): “Enhancing Water Security in the Middle East”. MENA Water Security Task Force, Al-Sharq Strategic Research, Al-Sharq Forum, Istanbul, Turkey; and Colorado School of Mines, Denver, CO, USA. Published on 20 March 2023. https://research.sharqforum.org/mena-water-security-task-force/
https://www.researchgate.net/publication/367190828_Chapter_4_Potential_Solutions_for_the_Water_Conflict_Between_Palestinians_and_Israelis

Salem, H.S. (2009). The Red Sea–Dead Sea Conveyance (RSDS) Project: A solution for some problems or a cause for many problems? In: Messerschmid, C., El-Jazairi, L., Khatib, I., Al Haj Daoud, A. (Eds.): The Conference Proceedings of Water: Values and Rights. United Nations Development Programme (UNDP), Programme of Assistance of the Palestinian People (PAPP), Palestinian Water Authority (PWA), and Palestine Academy for Science and Technology (PALAST). Ramallah, Palestine, April 13–15, 2009. PP: 300–366, 726p. https://www.palast.ps/en/publications/proceedings-1st-international-conference-water-values-and-rights

https://www.researchgate.net/publication/299563326_The_Red_Sea-Dead_Sea_Conveyance_RSDSC_Project_A_Solution_for_Some_Problems_or_A_Cause_for_Many_Problems

Salem, H.S. (2011a). Social, environmental and security impacts of climate change on the Eastern Mediterranean. In: Brauch, H.G., Spring, Ú.O., Mesjasz, C., Grin, J., Kameri-Mbote, P., Chourou, B., Dunay, P., Birkmann, J. (Eds.): Coping with Global Environmental Change, Disasters and Security – Threats, Challenges, Vulnerabilities and Risks. The Institute for Environment and Human Security, United Nations University (UNU-EHS). Hexagon Series on Human and Environmental Security and Peace, Berlin – Heidelberg – New York: Springer-Verlag. PP: 421–445. 1815p.

https://link.springer.com/chapter/10.1007/978-3-642-17776-7_23
https://www.researchgate.net/publication/299562984_Social_Environmental_and_Security_Impacts_of_Climate_Change_of_the_Eastern_Mediterranean

Salem, H.S. (2011b). Pollution of coastal areas on the Mediterranean Sea: The Gaza Strip as a case study – A real environmental threat and a big challenge. A paper presented at the Working Meeting of the MIRA Project Expert Group on Decontamination of the Mediterranean (WP7) “Mediterranean Innovation and Research Cooperation Action” (MIRA). Funded by the DG Research of the European Commission MIRA and FP7, Casablanca – Morocco, November 27–30, 2011.

https://www.researchgate.net/publication/319391700_Pollution_of_Coastal_Areas_on_the_Mediterranean_Sea_The_Gaza_Strip_As_a_Case_Study_-_A_Real_Environmental_Threat_and_a_Big_Challenge

Salem, H.S. (2019a). No sustainable development in the lack of environmental justice. Environmental Justice, 12(3-June):140–157.

https://doi.org/10.1089/env.2018.0040
https://www.liebertpub.com/doi/10.1089/env.2018.0040
https://www.researchgate.net/publication/342110072_No_Sustainable_Development_in_the_Lack_of_Environmental_Justice_Full_Paper

Salem, H.S. (2019b). Agriculture status and women’s role in agriculture production and rural transformation in the Occupied Palestinian Territories. Journal of Agriculture and Crops, 5(8-August):132–150.

https://doi.org/10.32861/jac.5(8)132.150 https://arpgweb.com/journal/journal/14

https://www.researchgate.net/publication/334770801_Agriculture_Status_and_Women's_Role_in_Agriculture_Production_and_Rural_Transformation_in_the_Occupied_Palestinian_Territories_Journal_of_Agriculture_and_Crops_2019_58_132-150

Salem, H.S. (2020). Geopolitical challenges, complexities, and future uncertainties in the Occupied Palestinian Territories: Land and population’s perspectives. New Middle Eastern Studies, 10(1):45–82. https://doi.org/10.29311/nmes.v10i1.3639

https://journals.le.ac.uk/ojs1/index.php/nmes/article/view/3639
https://www.researchgate.net/publication/344474119_Geopolitical_Challenges_Complexities_and_Future_Uncertainties_in_the_Occupied_Palestinian_Territories_Land_and_Population's_Perspectives

Salem, H.S., Isaac, J. (2007). Water agreements between Israel and Palestine and the region’s water argumentations between policies, anxieties and sustainable development. A keynote paper presented at The International Conference on Green Wars: Environment between Conflict and Cooperation in the Middle East and North Africa (MENA). Beirut, Lebanon, November 2–3, 2007. Sponsored by the Middle-East Office of the Heinrich Boell Foundation (HBF), Berlin, Germany.

http://www.afes-press.de/html/Report_Green%20Wars_%20Conference_Beirut_Nov2007.pdf
https://www.researchgate.net/publication/242313866_Water_Agreements_between_Israel_and_Palestine_and_the_Region's_Water_Argumentations_between_Policies_Anxieties_and_Unsustainable_Development

Salem, H.S., Yihdego, Y., Muhammed, H.H. (2021). The status of freshwater and reused treated wastewater for agricultural irrigation in the Occupied Palestinian Territories. Journal of Water and Health, 19(1-February):120–158. https://doi.org/10.2166/wh.2020.216
https://www.researchgate.net/publication/345063663_The_Status_of_Freshwater_and_Reused_Treated_Wastewater_for_Agricultural_Irrigation_in_the_Occupied_Palestinian_Territories_httpsdoiorg102166wh2020216

Salem, H.S, Yihdego, Y., Pudza, M.Y. (2022). Water strategies and water–food Nexus: Challenges and opportunities towards sustainable development in various regions of the world*. Sustainable Water Resources Management, 8:114:54p. [*Including the Gulf Corporation Council’s Countries, Central Asia Countries and the Caucasus, China, Africa, and Canada.] https://doi.org/10.1007/s40899-022-00676-3
https://www.researchgate.net/publication/361972621_Water_strategies_and_water-food_Nexus_challenges_and_opportunities_towards_sustainable_development_in_various_regions_of_the_World

Yihdego, Y., Salem, H.S., and Muhammed, H.H. (2019). Agricultural pest management policies during drought: Case studies in Australia and the State of Palestine. Natural Hazards Review, 2019, 20(1-February), 1–10. DOI:10.1061/(ASCE)NH.1527-6996.0000312. (Published by American Society of Civil Engineers, Reston (ASCE), Reston, VA, USA). https://ascelibrary.org/doi/abs/10.1061/%28ASCE%29NH.1527-6996.0000312 and https://www.researchgate.net/publication/342110444_Agricultural_Pest_Management_Policies_during_Drought_Case_Studies_in_Australia_and_the_State_of_Palestine_Full_Paper

Kind regards,

Prof. Dr. Hilmi S. Salem

Sustainable Development Research Institute

Bethlehem, West Bank, Palestine (Occupied)

6. PLOS authors have the option to publish the peer review history of their article (what does this mean?). If published, this will include your full peer review and any attached files.

Reviewer #1: No

---

## [Author Response · Author response to Decision Letter 0]

17 Jul 2023

Responses to all comments are listed in the attached document: Response to Reviewers.

---

## [Decision Letter · Decision Letter 1]

27 Sep 2023

PONE-D-22-25946R1Effect of agricultural extension on technology adoption by Palestinian farmers under Israeli occupation in the West BankPLOS ONE

Dear Dr. Tomoki,

Thank you for submitting your manuscript to PLOS ONE. After careful consideration, we feel that it has merit but does not fully meet PLOS ONE’s publication criteria as it currently stands. Therefore, we invite you to submit a revised version of the manuscript that addresses the points raised during the review process.

Kindly respond to all the comments and issues raised by the reviewer. If not clarify why you did not respond.

The following paper can help to strengthen the background of your manuscript.

Ali, E. (2021). Farm households’ adoption of climate-smart practices in subsistence agriculture: Evidence from northern Togo.  Environmental Management 67, 949–962. https://doi.org/10.1007/s00267-021-01436-3

Ali, E., Monkounti, Y (2020) Adoption of Biofeed technology in fighting against fruits fly in Togo. Food Systems 5, 157 - 180. DOI: 10.15122/isbn.978-2-406-11062-0.p.0157

We look forward to receiving your revised manuscript.

Kind regards,

Essossinam Ali, Ph.D

Academic Editor

PLOS ONE

Journal Requirements:

Additional Comments from the Editorial Staff:

One or more of the reviewers has recommended that you cite specific previously published works. Members of the editorial team have determined that the works referenced are not directly related to the submitted manuscript. As such, please note that it is not necessary or expected to cite the works requested by the reviewer.

Reviewers' comments:

Reviewer's Responses to Questions

**Comments to the Author**

1. If the authors have adequately addressed your comments raised in a previous round of review and you feel that this manuscript is now acceptable for publication, you may indicate that here to bypass the “Comments to the Author” section, enter your conflict of interest statement in the “Confidential to Editor” section, and submit your "Accept" recommendation.

Reviewer #2: (No Response)

2. Is the manuscript technically sound, and do the data support the conclusions?

Reviewer #2: Yes

3. Has the statistical analysis been performed appropriately and rigorously? 

Reviewer #2: Yes

4. Have the authors made all data underlying the findings in their manuscript fully available?

Reviewer #2: Yes

5. Is the manuscript presented in an intelligible fashion and written in standard English?

Reviewer #2: Yes

6. Review Comments to the Author

Reviewer #2: Please find the attached PDF file.

New comments on the improved manuscript: PONE-D-22-25946R1

Dated and Submitted on: 29 August 2023

Dear Respected Editor-in-Chief:

Dear Respected Authors:

First, it is my great pleasure to review the improved version of the Manuscript (Ms) re-submitted by scholar colleagues from Japanese universities. I appreciate their efforts.

Commenting on the “improved” manuscript re-submitted by the authors, please find below my recent (new-old) comments and kindly feel free to send this letter, with the comments below, to the authors, considering the fact that the authors have NOT responded positively to some of my comments given in the fist review, while they responded well on some other comments that I have made.

Accordingly, most of my comments below are with reference to my comments in the first review.

My final decision, based on the comments below, I support publishing the Ms in your Journal “Plos One”, if the authors consider the comments mentioned below. However, the final decision is the Journal’s Editorial team.

...........

Please see the Attached file.

7. PLOS authors have the option to publish the peer review history of their article (what does this mean?). If published, this will include your full peer review and any attached files.

Reviewer #2: **Yes: **Prof. Dr. Hilmi S. Salem

---

## [Author Response · Author response to Decision Letter 1]

28 Sep 2023

Thank you for your careful review of our manuscript and recommendation of the informative papers. We have incorporated all your comments into the manuscript, so please confirm it.

---

## [Editor Report · Decision Letter 2]

13 Oct 2023

PONE-D-22-25946R2Effect of agricultural extension on technology adoption by Palestinian farmers under Israeli occupation in the West BankPLOS ONE

Dear Dr. Nakamura Tomoki,

Thank you for submitting your manuscript to PLOS ONE. After careful consideration, we feel that it has merit but does not fully meet PLOS ONE’s publication criteria as it currently stands. Therefore, we invite you to submit a revised version of the manuscript that addresses the points raised during the review process.

Please review the abstract as suggested and send it back. Please check the Journal guide and comply with the reference style in the reference list.

We look forward to receiving your revised manuscript.

Kind regards,

Essossinam Ali, Ph.D

Academic Editor

PLOS ONE

Journal Requirements:

Additional Editor Comments :

Dear Dr. Nakamura Tomoki,

Thank you for revising your manuscript. Before proceeding with the publication, following comments should be addressed.

1. The abstract needs to be reviewed. There is no need for the first sentence. It can be deleted. Please give a brief description of the problem statement following the objectives, the methodology, and data used in the paper. Then talk about the results and their implication as policy recommendations.

2. The following papers can be helpful

Ali, E. (2021). Farm households’ decision of adoption of climate-smart practices in subsistence agriculture: Evidence from Northern Togo. Environmental Management, 67: 949–962 https://doi.org/10.1007/s00267-01436-3

Ali, E., Mounkounti, Y. (2020). Adoption de la technologie Biofeed dans la lutte contre la mouche des fruits au Togo. Sytèmes Alimentaires/ Food Systems, 5(1), 157-180. https://dx.doi.org/10.15122/isbn.978-2-406-11062-0.p.0157

Ali., E., Awade, E.N., Abdoulaye, T. (2020). Gender and impact of climate change adaptation on soybean farmers’ revenue in rural Togo, West Africa. Cogent Food & Agriculture 6(1): 1743625. https://doi.org/10.1080/23322039.2020.1783910

Comments from PLOS Editorial Office:

We note that the decision letter has recommendations to cite specific previously published works. As always, we recommend that you please review and evaluate the requested works to determine whether they are relevant and should be cited. It is not a requirement to cite these works. We appreciate your attention to this request.

---

## [Author Response · Author response to Decision Letter 2]

13 Oct 2023

All comments from the editor have been reflected into the manuscript. For details, please refer to the attached file: Response to Editor Comment.

I checked the Journal guide and modified the reference style in the reference list.

---

## [Editor Report · Decision Letter 3]

25 Oct 2023

Effect of agricultural extension on technology adoption by Palestinian farmers under Israeli occupation in the West Bank

PONE-D-22-25946R3

Dear Dr. Nakamura Tomoki,

We’re pleased to inform you that your manuscript has been judged scientifically suitable for publication and will be formally accepted for publication once it meets all outstanding technical requirements.

Kind regards,

Essossinam Ali, Ph.D

Academic Editor

PLOS ONE
---

## [Editor Report · Acceptance letter]

30 Oct 2023

PONE-D-22-25946R3 

Effect of agricultural extension on technology adoption by Palestinian farmers under Israeli occupation in the West Bank 

Dear Dr. Tomoki:

I'm pleased to inform you that your manuscript has been deemed suitable for publication in PLOS ONE. Congratulations! Your manuscript is now with our production department. 

Kind regards, 

on behalf of

Dr. Essossinam Ali 

Academic Editor

PLOS ONE